# Nanocurvature-induced field effects enable control over the activity of single-atom electrocatalysts

Bingqing Wang [1,7], Meng Wang[1,2,7], Ziting Fan [1], Chao Ma[4], Shibo Xi[5], Lo-Yueh Chang[6], Mingsheng Zhang[2], Ning Ling[1], Ziyu Mi[5], Shenghua Chen [4], Wan Ru Leow [5], Jia Zhang [3], Dingsheng Wang [4] & Yanwei Lum [1,2] ✉

Tuning interfacial electric fields provides a powerful means to control electrocatalyst activity. Importantly, electric fields can modify adsorbate binding energies based on their polarizability and dipole moment, and hence operate independently of scaling relations that fundamentally limit performance. However, implementation of such a strategy remains challenging because typical methods modify the electric field non-uniformly and affects only a minority of active sites. Here we discover that uniformly tunable electric field modulation can be achieved using a model system of single-atom catalysts (SACs). These consist of $M-N_4$ active sites hosted on a series of spherical carbon supports with varying degrees of nanocurvature. Using in-situ Raman spectroscopy with a Stark shift reporter, we demonstrate that a larger nanocurvature induces a stronger electric field. We show that this strategy is effective over a broad range of SAC systems and electrocatalytic reactions. For instance, Ni SACs with optimized nanocurvature achieved a high CO partial current density of ~400 mA cm$^{-2}$ at >99% Faradaic efficiency for $CO_2$ reduction in acidic media.

The successful development of electrochemical energy conversion technologies hinges on the ability to design electrocatalysts that can facilitate the desired cathodic and anodic reactions with high efficiency and activity[1]. Important examples of these include water splitting[2–7], fuel cell reactions[8–12] and $CO_2$ reduction[13–17]. Such reactions are important as they offer a route to realize net-zero-emission production of value-added chemicals/fuels and on-demand generation of electricity[18].

Tuning interfacial electric fields potentially provides a powerful means to control electrocatalyst activity for a broad range of

reactions[19–27]. Electric fields can influence the catalytic rate by modifying adsorbate binding energies based on their polarizability and dipole moment[28–31]. This means that they operate independently of conventional scaling relations which place a fundamental limit on electrocatalyst performance[32,33]. For instance, Sargent and co-workers assembled Au nanoneedle catalysts, and an intense local electric field was concentrated at the tips, where the curvature is highest[34]. It was proposed that the high local electric field at the nanoneedle tip can concentrate cations, which then stabilizes intermediates and lowers reaction pathway energetics[34,35]. As a result, the Au nanoneedle

[1]Department of Chemical and Biomolecular Engineering, National University of Singapore, Singapore 117585, Republic of Singapore. [2]Institute of Materials Research and Engineering (IMRE), Agency for Science, Technology and Research (A*STAR), 2 Fusionopolis Way, Innovis #08-03, Singapore 138634, Republic of Singapore. [3]Institute of High Performance Computing (IHPC), Agency for Science, Technology, and Research (A*STAR), 1 Fusionopolis Way, #16-16 Connexis, Singapore 138632, Republic of Singapore. [4]Department of Chemistry, Tsinghua University, Tsinghua, China. [5]Institute of Sustainability for Chemicals, Energy and Environment (ISCE2), Agency for Science, Technology and Research (A*STAR), 1 Pesek Road, Singapore 627833, Republic of Singapore. [6]National Synchrotron Radiation Research Centre, Hsinchu, Taiwan. [7]These authors contributed equally: Bingqing Wang, Meng Wang. ✉e-mail: lumyw@nus.edu.sg

catalysts achieved high selectivity and activity towards $CO_2$ reduction to CO. Besides this tip enhancement effect, various groups have found that the alkali metal cation used in the electrolyte (e.g. $Li^+$, $K^+$) can also have an effect on interfacial electric fields, which in turn impacts catalyst activity and selectivity[20,29].

Although important progress has been made by these previous works, further realizing the potential of electric fields for electrocatalysis remains challenging for several reasons. (1) The typical strategy employed is through the construction of sharp tipped structures, which means that electric field modulation does not occur uniformly throughout the catalyst surface and only affects a minority of active sites. (2) It is not known whether a stronger electric field necessarily enhances activity, or if there exists an optimal value that depends on the reaction of interest and electrocatalyst system used. (3) Besides finite-element numerical method simulations, there has been no experimental method that has directly evaluated the effect of curvature on the interfacial electric field under electrochemical conditions. Addressing these issues could hence aid the development of rational design rules for manipulating the interfacial electric field and understanding of its effects on electrocatalyst activity and selectivity.

Here we show that uniformly tunable electric field modulation can be achieved using a model system of single-atom catalysts (SACs). Such catalysts typically consist of isolated metal atoms that are anchored and singly dispersed onto a substrate such as a carbon-based support[36-43]. In this work, our SAC system consists of $M-N_4$ active sites hosted on a series of spherical carbon supports with varying degrees of nanocurvature. By using in-situ Raman spectroscopy with $SCN^-$ as the vibrational Stark effect electric field reporter, we further experimentally demonstrate that a higher degree of nanocurvature does indeed induce a stronger interfacial electric field. Hence, the nanocurvature of the carbon support can be tuned to exert control over the SAC activity.

We applied this strategy to Ni, Fe and Co SACs and discovered that nanocurvature significantly impacts their activity for a broad range of reactions, especially for those with reaction intermediates that feature a high dipole moment or polarizability. We first investigated electrochemical $CO_2$ reduction (CO$_2$R) in acidic media, and the Ni-SAC with the optimal nanocurvature exhibited a high CO partial current density of ~400 mA/cm$^2$ at >99% Faradaic efficiency. This was further extended to alkaline oxygen reduction reaction (ORR), where the best Fe SAC displayed a 0.91 V half-wave potential in rotating disk electrode (RDE) measurements. We also show that nanocurvature can control the activity for the oxygen evolution reaction (OER) and hydrogen evolution reaction (HER) in alkaline media. These results contrast with HER in acidic media, where we do not observe any effect of nanocurvature on SAC activity. This is because the *H intermediate has no dipole moment or polarizability[29,30] and hence its binding energy remains unaltered by interfacial electric fields.

## Results

We began by exploring the effect of interfacial electric fields on intermediate adsorption energies using density functional theory (DFT) simulations. Focusing first on $CO_2$R to CO on the commonly reported Ni-$N_4$ SAC system[44,45], we calculated the Gibbs free energy changes in the reaction pathway without the influence of an electric field (Fig. 1a). For consistency, we included a $K^+$ cation and solvent water into the simulation models (Supplementary Fig. 1), similar to prior literature reports[21,34]. Following the methodology of Norskov and co-workers[19], the Gibbs free energy of each reaction stage (*CO$_2$, *COOH and *CO) was then computed as a function of electric field strength (Supplementary Fig. 2). Based on these results, we observed that an optimized electric field results in lowering of the free energy change of the potential-determining step *CO$_2$ → *COOH (Fig. 1a). For example, the free energy change for this step (limiting potential) is reduced from 0.96 eV to 0.88 eV when an electric field of −0.4 VÅ$^{-1}$ is applied, but increased to 0.9 eV when the electric field was further

increased to −0.8 VÅ$^{-1}$. In addition, the limiting potential difference between CO$_2$R and HER ($U_L(CO_2)$-$U_L(H_2)$; $U_L = -\Delta G_0/e$) was also calculated and employed as the descriptor for CO selectivity[46,47], with a more positive value of $U_L(CO_2)$-$U_L(H_2)$ indicating a higher CO$_2$R selectivity. These findings are summarized in Fig. 1b, where an optimal electric field strength was observed for this descriptor with Ni-$N_4$. In addition, similar trends were also found for Fe-$N_4$ and Co-$N_4$ active sites (Supplementary Fig. 3, 4)

Next, we expanded the DFT study to ORR on the Fe-$N_4$ SAC system[48,49]. Similarly, we computed the Gibbs free energy change of each reaction stage (*OOH, *O and *OH) as a function of electric field strength (Supplementary Fig. 5). Figure 1c shows the reaction pathway free energy change with and without an electric field. Likewise, we find that the free energy change of the potential-determining step is reduced from 0.59 eV to 0.43 eV when an electric field of −1.2 VÅ$^{-1}$ is applied. Figure 1d shows a contour plot of the ORR thermodynamic limiting potential as a function of $\Delta G_{OH}$ and $\Delta G_{OOH}$, as previously described by Norskov and co-workers[50]. On this plot, we superimpose our calculated values for the Fe-$N_4$ SAC system with (white triangles) and without (black triangle) an electric field. In addition, the well-known scaling relationship between $\Delta G_{OH}$ and $\Delta G_{OOH}$ for SAC systems is also shown (black dotted line) (Supplementary Fig. 6). Here, we clearly see that electric field modulation of intermediate binding energies can operate independently of scaling relations and shift $\Delta G_{OH}$ and $\Delta G_{OOH}$ towards more optimum values for ORR.

In sum, our DFT results indicate that electric field modulation can be used as a knob to control electrocatalyst activity and bypass conventional scaling relations. Motivated by these findings, we sought to experimentally implement this electric field modulation concept. To do this, we envisioned constructing a model system consisting of M-$N_4$ single-atom sites hosted on nanosized spherical carbon supports. We reasoned that varying the carbon sphere diameter would enable control over the degree of nanocurvature, with a smaller sphere diameter resulting in a higher nanocurvature.

Based on this, we used a finite-element numerical method (see Supplementary Note 1) to qualitatively understand the impact of nanocurvature on interfacial electric fields and provide an indication of the desired sphere diameter range to synthesize for our experiments[21,34]. Consistent with our expectations, the simulation results in Fig. 1e, f and Supplementary Fig. 7 show that as the sphere diameter decreases from 1000 nm to 100 nm, the interfacial electric field indeed increases by a factor of ~22. These results demonstrate that the interfacial electric field can in principle be modulated by tuning the sphere diameter and this concept is illustrated in the schematic in Fig. 2a. In addition, the enhancement in the interfacial electric field leads to a concomitant rise in the counter-ion ($K^+$) concentration near the catalyst surface[34] (Supplementary Fig. 8). The electrostatic effect experienced by adsorbed intermediates would thus be a combination of these positively charged counter-ions and the negatively charged electrode. The resulting interfacial electric field then alters adsorbate binding energies according to the equation[19]:

$$G_{ads} = G_{ads}^{PZC} + \mu \vec{E} - \frac{\alpha}{2} \vec{E}^2 \qquad (1)$$

$G_{ads}$ refers to the adsorbate binding energy, $G_{ads}^{PZC}$ refers to the adsorbate binding energy with no applied interfacial electric field, $\mu$ is the adsorbate dipole moment, $\alpha$ is the adsorbate polarizability and $\vec{E}$ is the interfacial electric field.

Encouraged by these results, we sought to realize this model system experimentally. We first synthesized hollow polymer nanospheres of varying diameters, following the procedure by Schüth[51] and co-workers with some modifications (see methods section for full details, Supplementary Figs. 9 and 10 and Supplementary Table 1). The

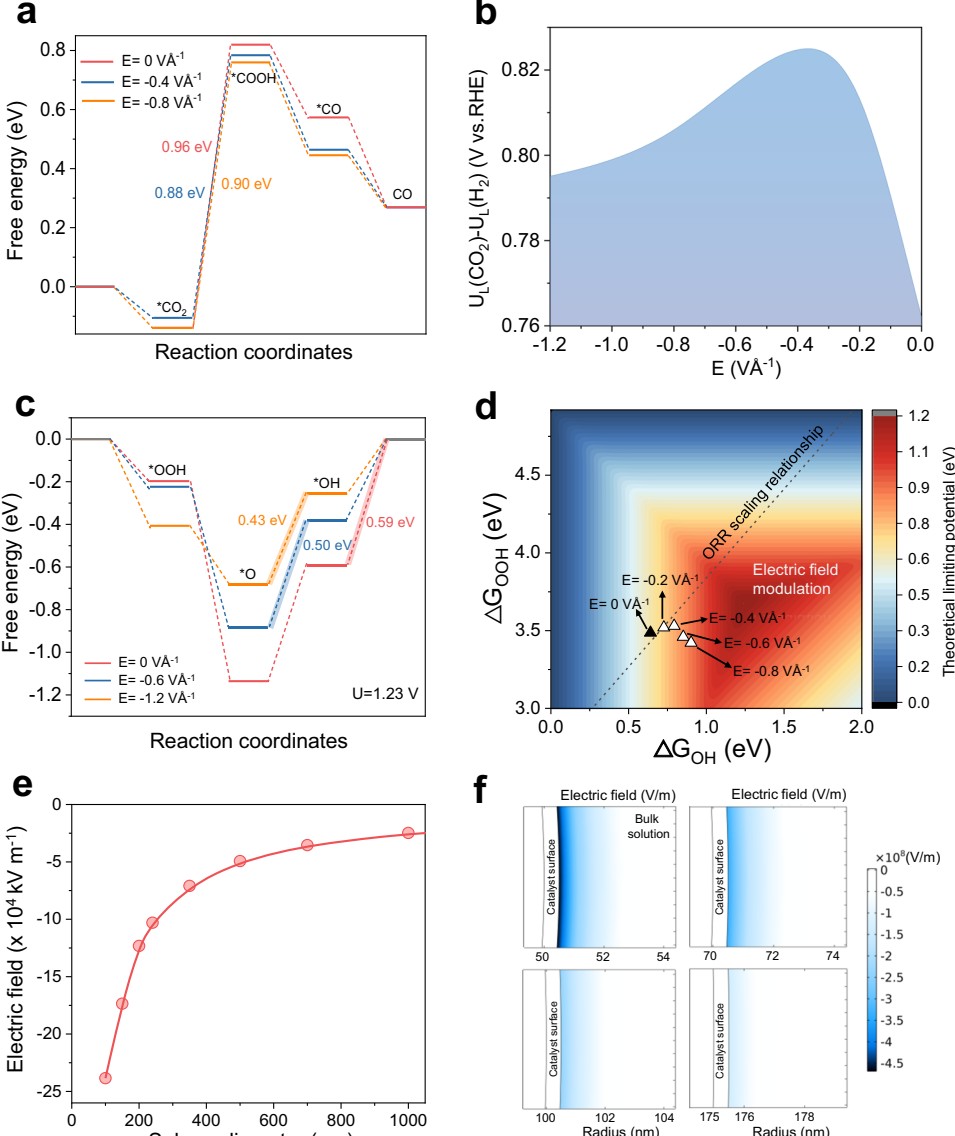

**Fig. 1 | Density functional theory and finite-element numerical method simulations. a** Reaction pathway Gibbs free energy diagrams and (**b**) theoretical limiting potential difference of $U_L(CO_2)$-$U_L(H_2)$ on Ni-N$_4$ as a function of interfacial electric field strength. **c** Reaction Gibbs free energy diagrams of ORR on Fe–N$_4$ under the influence of various electric field strengths. **d** Contour plot of the ORR limiting potential as a function of $\Delta G_{OH}$ and $\Delta G_{OOH}$. The values for Fe-SAC are shown with (white triangles) and without (black triangle) the influence of an electric field.

**e** Finite-element numerical method simulation results of the interfacial electric field strength as a function of sphere diameter. See Supplementary Note 1 for more details. **f** Computed electric field intensity distribution for 4 sphere diameters of 100, 140, 200 and 350 nm with additional results available in Supplementary Fig. 7. Note: the sphere radius rather than diameter is shown in the figure. Source data are provided as a Source Data file.

SACs were then created by mixing the appropriate metal salt to the hollow polymer nanospheres, followed by pyrolysis at high temperature in a tube furnace with dicyandiamide as the nitrogen source (Supplementary Fig. 11). The high temperature results in carbonization of the hollow polymer nanospheres, transforming them into hollow carbon nanospheres. At the same time, this process forms M–N$_4$ sites on the carbon nanospheres. In this work, we investigated the electric field modulation concept with 3 common types of SACs, namely Ni, Fe and Co for a broad range of electrocatalytic reactions. The SACs will be referred to as "(M)-SAC-(X)", where (M) is the identity of the transition metal atom and (X) is the sphere diameter in nm.

The synthesized Ni SACs of different sphere diameters were first examined using scanning electron microscopy (SEM) and transmission electron microscopy (TEM). SEM images show that the obtained Ni SACs exhibit the same spherical morphology and are of similar size to

the original polymer nanospheres. We statistically analyzed 100 particles based on these SEM images and average diameters obtained are 70 nm, 130 nm, 250 nm and 350 nm (Supplementary Fig. 12). Figure 2b–e shows the TEM images of the same catalysts, namely Ni-SAC-70, Ni-SAC-130, Ni-SAC-250 and Ni-SAC-350. Similarly, statistical analysis of the size distribution is included as an inset in each of the TEM images. In addition, the hollow nature of these SACs can be clearly observed, and we note the absence of any visible Ni nanoparticles or clusters (Supplementary Fig. 13).

Our X-ray diffraction (XRD) results also indicate the absence of any metallic Ni phases, with only 2 broad peaks associated with graphitic carbon for all Ni SACs (Supplementary Fig. 14). Aberration-corrected high-angle annular dark-field scanning transmission electron microscopy (AC-HAADF-STEM) was also performed, with Fig. 2f and Supplementary Fig. 15 showing the uniformly dispersed Ni single

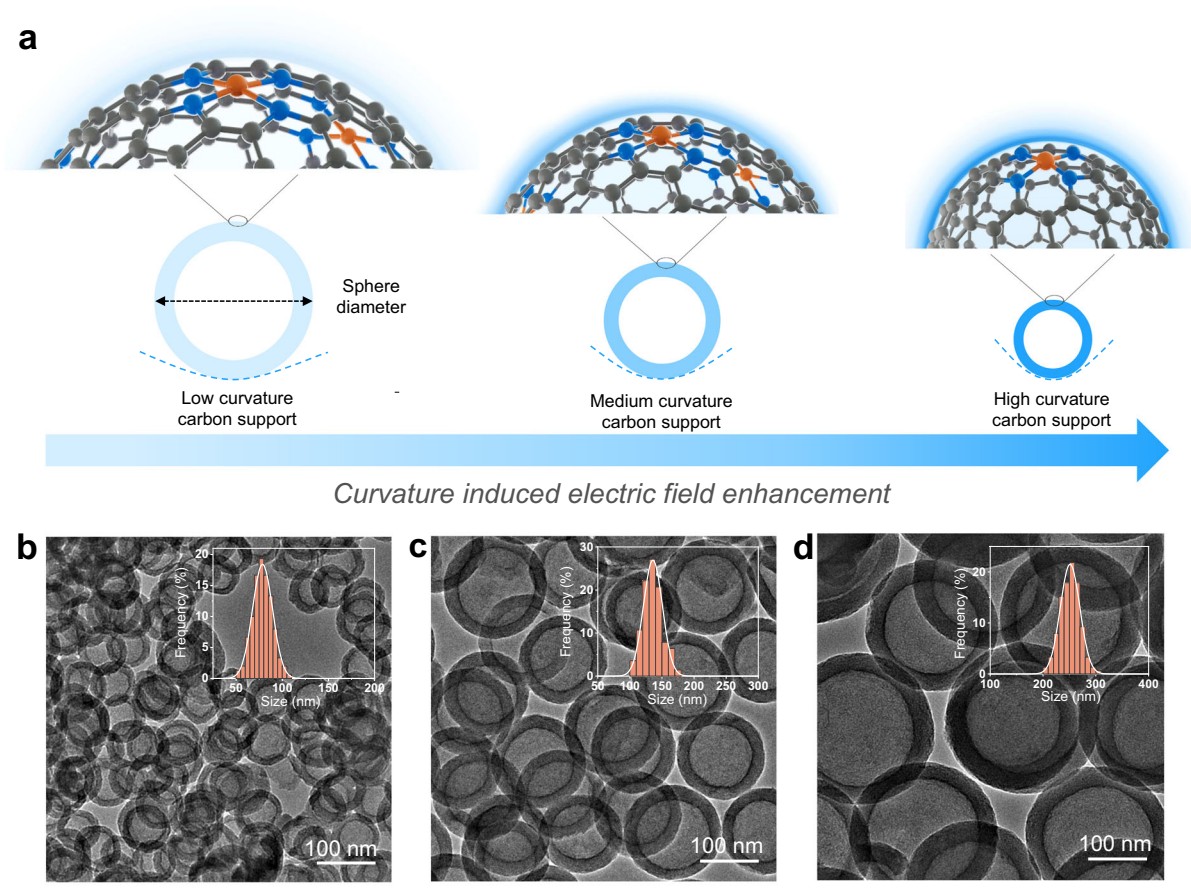

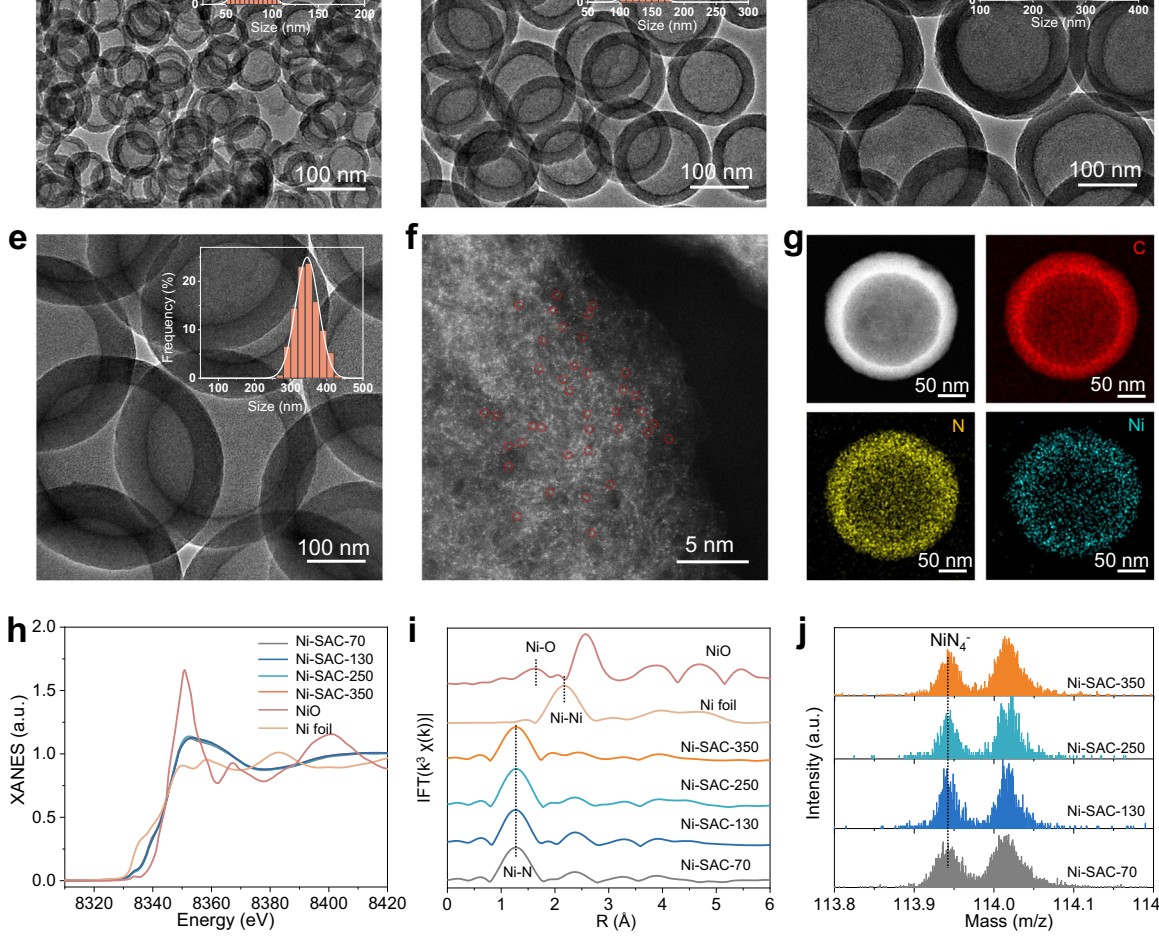

**Fig. 2 | Catalyst design principle and materials characterization. a** Schematic of our design principle, whereby modulation of the interfacial electric field can be achieved by the carbon support sphere diameter. **b**−**e** TEM images of Ni SACs, consisting of Ni single-atoms supported on carbon nanospheres of varying diameter. **b** Ni-SAC-70, (**c**) Ni-SAC-130, (**d**) Ni-SAC-250 and (**e**) Ni-SAC-350. The inset of each image shows the size distribution of the carbon nanospheres. More TEM images are shown in Supplementary Fig. 13. **f** AC-HAADF-STEM image of Ni-SAC-250. **g** HAADF-STEM and corresponding EDS mapping images of Ni-SAC-250. **h** Ni K-edge XANES spectra and **i** Ni K-edge Fourier-transformed (FT) $k^2$-weighted $\chi(k)$ functions of Ni-SACs with different sphere diameters. NiO and Ni foil were used as references. **j** TOF-SIMS high-resolution negative-ion spectra for Ni-SAC-70, Ni-SAC-130, Ni-SAC-250 and Ni-SAC-350. Source data are provided as a Source Data file.

atoms in the Ni SACs. Energy-dispersive X-ray spectroscopy (EDS) mapping shows that the elements C, N and Ni are uniformly distributed throughout each of the catalysts (Fig. 2g and Supplementary Figs. 16–18). The Ni metal loading in each Ni SAC was also quantified using inductively coupled plasma-atomic emission spectrometry (ICP-OES). The results show that all SACs have similar Ni metal loadings of ~0.81 wt.% (Supplementary Table 2).

Next, we used several spectroscopy characterization techniques to probe the electronic properties and local coordination structures of these Ni SACs. Raman spectroscopy measurements show similar D/G ratios of around 1.10, which indicates that they all have similar degrees of graphitization (Supplementary Fig. 19). X-ray photoelectron spectroscopy (XPS) confirms the presence of C, N and Ni elements in all the samples (Supplementary Fig. 20). The N 1s spectra of each Ni SAC was also deconvoluted into four peaks: pyridinic, pyrrolic, graphitic N and oxidized N. We find that the ratio of the different N species is similar for all SACs, regardless of the sphere diameter. The binding energy of Ni $2p$ in the Ni SACs are all located at ~854.8 eV. This indicates a higher oxidation state compared to metallic Ni (852.6 eV) but is lower than that of $Ni^{2+}$ (855.7 eV)[52].

X-ray absorption near-edge structure (XANES) spectra at the Ni K-edge further shows that the absorption edge of all four Ni-SAC samples is similar and are all located between Ni metal and NiO (Fig. 2h). This suggests that the oxidation state of Ni in these SACs is between $Ni^0$ and $Ni^{2+}$, in agreement with XPS results. Fourier transform (FT) extended X-ray absorption fine structure spectrometry (EXAFS) of the Ni SACs shows a dominant peak located at 1.3 Å, which can be attributed to the first coordination shell of Ni-N. We note the absence of a peak belonging to the second shell Ni-Ni interaction, and this confirms that Ni is dispersed as single-atoms rather than clusters, dual-atoms, or triple-atoms in the SACs (Fig. 2i). Furthermore, since the XAS data reveals that there are no major differences in the electronic structure of all four Ni-SAC samples, we infer that their magnetic spin states should be similar as well.

Wavelet transform analysis further confirms the atomic dispersion of Ni in all the samples, in which no Ni-Ni interactions are observed but only an intensity maximum attributed to Ni-N/C (Supplementary Fig. 21). In addition, the least-squares EXAFs curve fitting shows that the coordination number of the first coordination sphere of Ni in all the SACs are close to 4 (Supplementary Figs. 22, 23 and Supplementary Table 3). This indicates that the Ni species in these SACs exist predominantly as $Ni-N_4$, which is also confirmed by time-of-flight secondary ion mass spectrometry (TOF-SIMS) analysis (Fig. 2j)[53]. The double-layer capacitance of all the samples was further determined using cyclic voltammetry in a non-faradaic potential window. The results show no significant difference in the double-layer capacitance, which means that they possess similar electrochemical active surface areas (Supplementary Fig. 24). Kelvin probe atomic force microscopy (Fig. 3a) was also performed, and this confirmed that electric fields are higher for smaller sphere diameters (larger nanocurvature).

Taken together, our materials characterization results demonstrate that our experimental methodology allows us to construct a model system of Ni-SACs, with $Ni-N_4$ active sites hosted on hollow spherical carbon supports of varying diameters. Importantly, all SACs possess identical active sites and spherical morphology, which allows us to unambiguously study and investigate the impact of nanocurvature on SAC activity. Furthermore, similar methodologies were employed to prepare and characterize Co and Fe SACs of varying carbon sphere diameter, with similar outcomes obtained (Supplementary Figs. 25–40 and Supplementary Tables 2, 4 and 5). We also note that nanocurvature-induced strain effects[54–57] are unlikely to operate in our investigated SAC systems, which requires a much higher curvature of the support. More importantly, our XPS and XAS results do not indicate any significant changes in the electronic structure of our catalysts as a function of sphere diameter, which indicates that any nanocurvature-induced strain is negligible.

Next, we sought to take a step beyond previous studies by devising a new strategy to experimentally evaluate interfacial electric fields under electrochemical conditions with an applied potential. In the presence of an electric field, a molecule experiences perturbation of the energy of its vibrational modes and this is known as the vibrational Stark effect[58]. This can be probed using in-situ Raman spectroscopy, with stronger interfacial electric fields resulting in larger peak shifts. The magnitude of the peak shift is known to vary linearly with

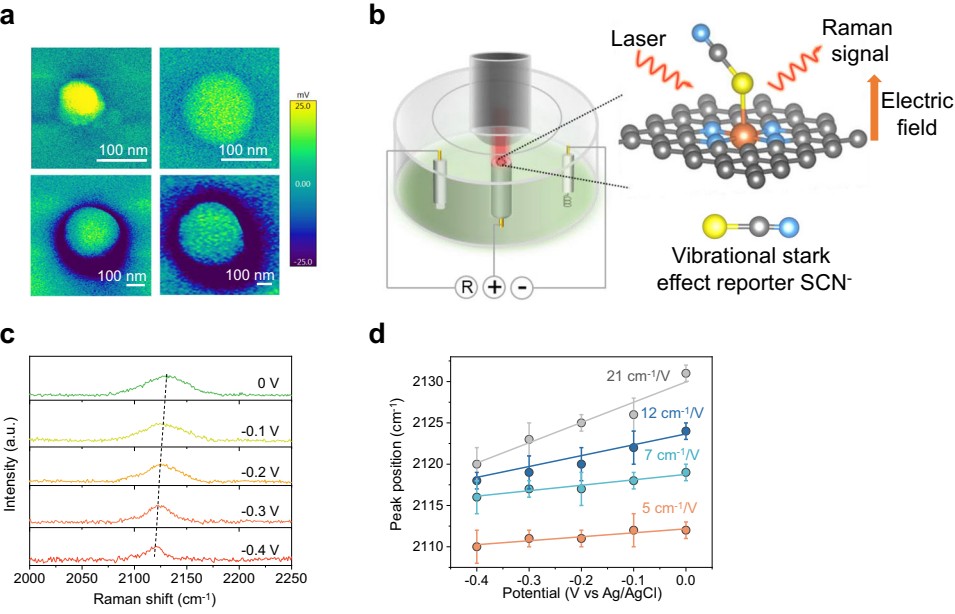

**Fig. 3 | Characterization of the electrocatalyst electric field. a** Electric field distribution on different size Ni-SACs deduced using Kelvin probe atomic force microscopy. **b** Schematic of probing the electric field of the active site in different sized Ni-SACs by using the vibrational stark effect. **c** Potential-dependent SHINERS spectra of SCN⁻ adsorbed on Ni-SAC-70 in 0.1 M NaClO₄ solution. **d** Peak positions of $\nu$SCN⁻-NiN₄ as a function of the electrode potential. All the error bars represent standard deviation based on three independent tests. Source data are provided as a Source Data file.

the applied potential, and the value of the slope is known as the Stark tuning rate[59]. A higher Stark tuning rate hence corresponds to a larger interfacial electric field.

For these experiments, we selected SCN⁻ as the electric-field Stark shift reporter because the CN stretching mode is known to be decoupled from other vibrational modes and highly localized among a pair of atoms[59–61]. SCN⁻ is also known to bind strongly to transition metal atoms, hence is typically used to poison SAC active sites[62]. More importantly, this means that SCN⁻ will selectively bind to the Ni single atoms and hence will directly report on the interfacial electric field experienced by the active sites of our Ni-SACs (Fig. 3b). Under normal in-situ Raman spectroscopy conditions, we were unable to detect the SCN⁻ peak. Hence, we employed shell-isolated nanoparticle-enhanced Raman spectroscopy (SHINERS) to enhance the Raman signal[63], which uses $SiO_2$-coated Au nanoparticles (see Supplementary Figs. 41 and 42 and methods section for more details).

Figure 3c shows representative in-situ Raman spectroscopy results for Ni-SAC-70 in 0.1 M $NaClO_4$ electrolyte containing 0.1 M KSCN, and we observe clear shifts in the SCN⁻ peak as a function of applied potential. Similar experiments were carried out with the other Ni-SACs (Supplementary Fig. 43), and from this we plotted the peak position as a function of applied potential (Fig. 3d). For all SACs, we observe linear relationships between the peak position and applied potential, and from this we calculated the Stark tuning rate (slope) for each case. We find that the Stark tuning rate increases with smaller sphere diameter (larger nanocurvature). For example, Ni-SAC-70 has a Stark tuning rate of 21 $cm^{-1}$/V as compared to only 5 $cm^{-1}$/V with Ni-SAC-350. Hence, we can conclude that the active sites on Ni-SAC-70 experience a larger interfacial electric field as compared to Ni-SAC-350. Importantly, this confirms our hypothesis that tuning the nanocurvature of the carbon support can indeed be used to control the interfacial electric field experienced by the SAC active site under electrochemical conditions.

Next, we investigated the impact of SAC nanocurvature on electrocatalysis, focusing on reactions which are known to have intermediates that possess a strong dipole moment and/or polarizability. This makes them susceptible to being influenced by interfacial electric fields. Hence, we selected $CO_2R$ as the first example, and performed our experiments in pH 1 acidic electrolyte (0.1 M $K_2SO_4$ + $H_2SO_4$) with the Ni-SACs. $CO_2R$ experiments were carried out in a custom-made electrochemical flow cell using gas diffusion electrodes, of a similar design to that previously reported in the literature[64] (Supplementary Fig. 44, see methods section for more details).

We performed $CO_2R$ measurements at constant current density from 50–400 mA $cm^{-2}$, with Fig. 4a showing the CO Faradaic efficiency (FE) results. We observed that Ni-SAC-70 obtained the highest CO FE of 95.0% at 50 mA $cm^{-2}$, but decreases significantly with higher current density, with similar trends observed with Ni-SAC-130. On the other hand, while Ni-SAC-250 exhibits a CO FE of only 90.0% at 50 mA $cm^{-2}$, it outperforms the SACs at the higher current densities with near-unity FE to CO (>99%). In-situ attenuated total reflection-infrared spectroscopy (ATR-IR) of different diameter Ni-SACs also indicates that the Ni-SAC-250 catalyst is better at converting $CO_2$ to CO (Supplementary Figs. 45 and 46). From these results, we show that sphere diameter does indeed have a significant impact on catalyst selectivity, with Ni-SAC-250 having the optimal nanocurvature and hence interfacial electric field for $CO_2R$ in acidic electrolyte (Supplementary Tables 6–9). Notably, Ni-SAC-250 achieves a CO partial current density of ~400 mA $cm^{-2}$ at >99% Faradaic efficiency and outperforms other similar electrocatalysts previously reported in the literature (Supplementary Table 10).

To understand if this concept could be extended to other transition metals, similar $CO_2R$ experiments in acidic electrolyte were performed with both Fe and Co SACs of varying sphere diameters. For Fe SACs, five different sphere diameters were tested: Fe-SAC-70, Fe-SAC-

130, Fe-SAC-250, Fe-SAC-350 and Fe-SAC-450 (Supplementary Tables 11–14). The FE results in Fig. 4b show that Fe-SAC-350 consistently displays the highest CO FE for all tested current densities, with a peak value of 88.5% at 100 mA $cm^{-2}$. As for Co SACs, six different sphere diameters were tested as well: Co-SAC-70, Co-SAC-130, Co-SAC-250, Co-SAC-350, Co-SAC-450 and Co-SAC-600 (Supplementary Table 15–18). From the results (Fig. 4c), the highest CO FE was observed for Co-SAC-450 across all current densities, with a peak value of 56.0% at 100 mA $cm^{-2}$.

To directly compare the CO turnover frequency (TOF) for all the SACs, we performed $CO_2R$ with each of them at a constant potential of −1.5 V vs RHE. Based on the obtained FE and current density, we calculated the TOF based on the assumption that 100% of the metal atoms exist as single-atoms and can serve as active sites for CO generation (see Supplementary Note 2 for calculation details). From the data in Fig. 4d, we observe a volcano relationship between the TOF and sphere diameter for all three transition metal SAC systems, with Ni-SAC-250, Fe-SAC-350 and Co-SAC-450 having the highest values for each case. For example, Ni-SAC-250 has a TOF value of $2.52 \times 10^4$ $h^{-1}$, which is 4.2 times higher than that of Ni-SAC-70. Such a volcanic relationship is expected since there should exist an optimal value of the interfacial electric field where reaction energy barriers between all the reaction intermediates are the lowest, consistent with our DFT calculation results (Fig. 1b).

The local chemical environment has been previously indicated as a factor that affects the $CO_2R$ performance[65,66]. To investigate its role in influencing the activity of our Ni-SAC catalyst, we prepared electrodes with different catalyst loadings of 0.2, 0.5 and 1.0 mg $cm^{-2}$ for each of the Ni-SACs to study the role of the local chemical environment. This is because each loading would be different in terms of catalyst packing, diffusion rates and microenvironments (local pH and $CO_2$ availability). Supplementary Fig. 47 shows that regardless of the catalyst loading and the bulk electrolyte concentration, the optimal sphere diameter is 250 nm (Ni-SAC-250). Based on this result, we can conclude that the local chemical environment does not control the optimal sphere diameter for the catalyst. Rather, this sphere diameter gives the best performance because its nanocurvature gives rise to an optimal interfacial electric field strength for $CO_2R$. In addition, we also ruled out the effect of shell thickness variation between the different diameter catalysts. This was because we found that catalysts with the same sphere diameter (250 nm), but different shell thicknesses exhibit similar $CO_2R$ performance (Supplementary Figs. 48 and 49).

Interestingly, we also note that each transition metal SAC system (Fe/Co/Ni) has a different optimal sphere diameter where the best FE and TOF is obtained. This is reasonable, since the binding energies of reaction intermediates are not expected to be the same for each transition metal SAC system, hence the optimal interfacial electric field should be different as well. Importantly, these experiments confirm that nanocurvature can be used to control the SAC activity and selectivity for $CO_2R$ in acidic media.

In addition, we performed $CO_2R$ for 25 h of continuous operation with the best performing Ni-SAC-250 catalyst at 200 mA $cm^{-2}$. The results in Fig. 4e show stable performance for CO generation over this period, with an FE of ~97% in the initial 3 h. A slight decay in the CO FE to 93% is observed near the end of the period, which we attribute to gradual flooding of the gas diffusion layer, rather than catalyst deactivation[67]. Post-experiment characterization of Ni-SAC-250 was also carried out with TEM, AC-HAADF-STEM and XAS (Supplementary Fig. 50). From these results, we observed no significant changes in the structure, morphology and Ni−$N_4$ active sites even after operation over this extended 25-h duration. This is also consistent with our DFT calculation result that the catalyst is stable under an applied electric field (Supplementary Fig. 51)

Our next objective was to determine if our concept could also be extended to influence a broad range of electrocatalytic reactions. As

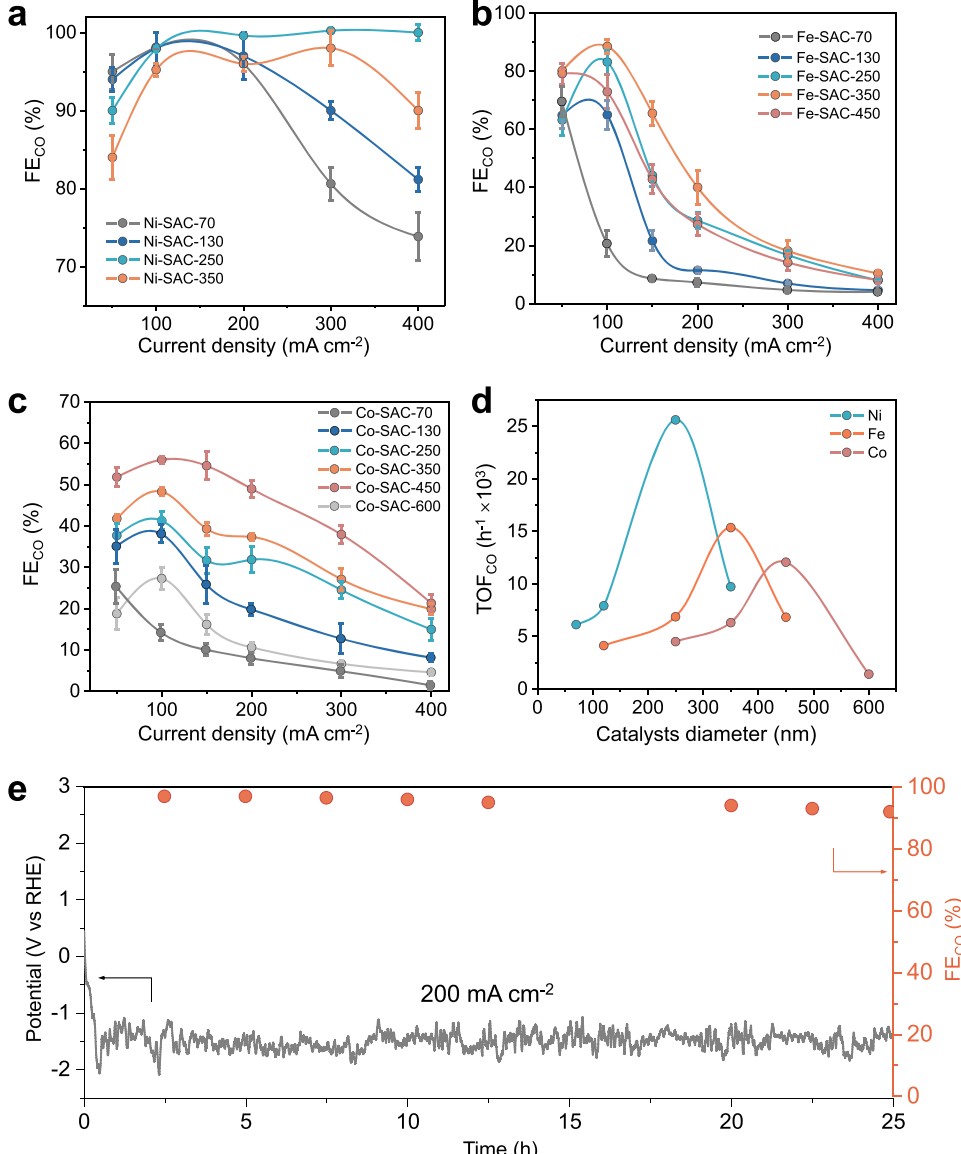

**Fig. 4 | CO₂R performance of the SACs.** FE for CO production over (**a**) Ni-SACs, (**b**) Fe-SACs and (**c**) Co-SACs with different diameters as a function of applied current density measured in acidic electrolyte (0.1 M K₂SO₄ + H₂SO₄). All the error bars represent standard deviation based on three independent samples. **d** TOFs for CO production at −1.5 V vs RHE over Ni-SACs, Fe-SACs, and Co-SACs as function of SAC sphere diameter. **e** Stability test of Ni-SAC-250 catalyst at 200 mA cm⁻² measured in acidic electrolyte. Source data are provided as a Source Data file.

such, we investigated the alkaline ORR, HER and OER reactions with our transition metal SAC systems. This selection comprises a good representation of important electrocatalytic oxidation and reduction reactions.

For alkaline ORR, we selected the Fe SAC system because catalysts with Fe−N₄ active sites are well-known to be highly active for this reaction[48,49]. Linear sweep voltammetry (LSV) was performed in O₂ saturated 0.1 M KOH electrolyte using a rotating disk electrode at 1600 rpm. Fe-SACs with diameters of 70, 130, 250 and 350 nm were tested. The LSV results in Fig. 5a show that Fe-SAC-70 with the smallest sphere diameter achieves the highest activity among the four Fe-SACs. Figure 5b summarizes the LSV results, with Fe-SAC-70 showing the highest half-wave (E₁/₂) potential of 0.91 V and the highest ORR kinetic current density (jₖ) of 24.9 mA cm⁻² at 0.85 V. In addition, the activity also appears to decrease with increasing sphere diameter.

We also observed that the sample with optimal nanocurvature (Fe-SAC-70) is different from that with acidic CO₂R (Fe-SAC-350). This is to be expected because ORR intermediates and reaction pathways are

very different from that of CO₂R. Furthermore, it is likely that the interfacial electric field changes significantly when the SACs are exposed to a different electrolyte. In addition, we do not observe a volcanic relationship between the SAC activity and sphere diameter. This is likely because the peak sphere diameter is smaller than Fe-SAC-70, which we found difficult to attain using our current SAC synthesis strategy.

Notably, we find that the ORR performance of Fe-SAC-70 is amongst the best as compared to similar SAC systems reported in the literature (Supplementary Table 19). Given this exceptional performance, a liquid Zn-air battery (ZAB) was assembled using Fe-SAC-70 as the air cathode (Supplementary Fig. 52). The Fe-SAC-70 shows an open-circuit voltage (OCV) of 1.45 V and delivers a maximum power density of 269 mW cm⁻², much higher than that of Fe-SAC-350 (132 mW cm⁻²) and surpassing other reported nitrogen-doped carbon supported metal-based SACs (Supplementary Table 20). In addition, the Fe-SAC-70 based ZAB cathode also exhibits a much higher specific capacity (810 mAh g⁻¹ Zn) than a ZAB using a benchmark Pt/C catalyst

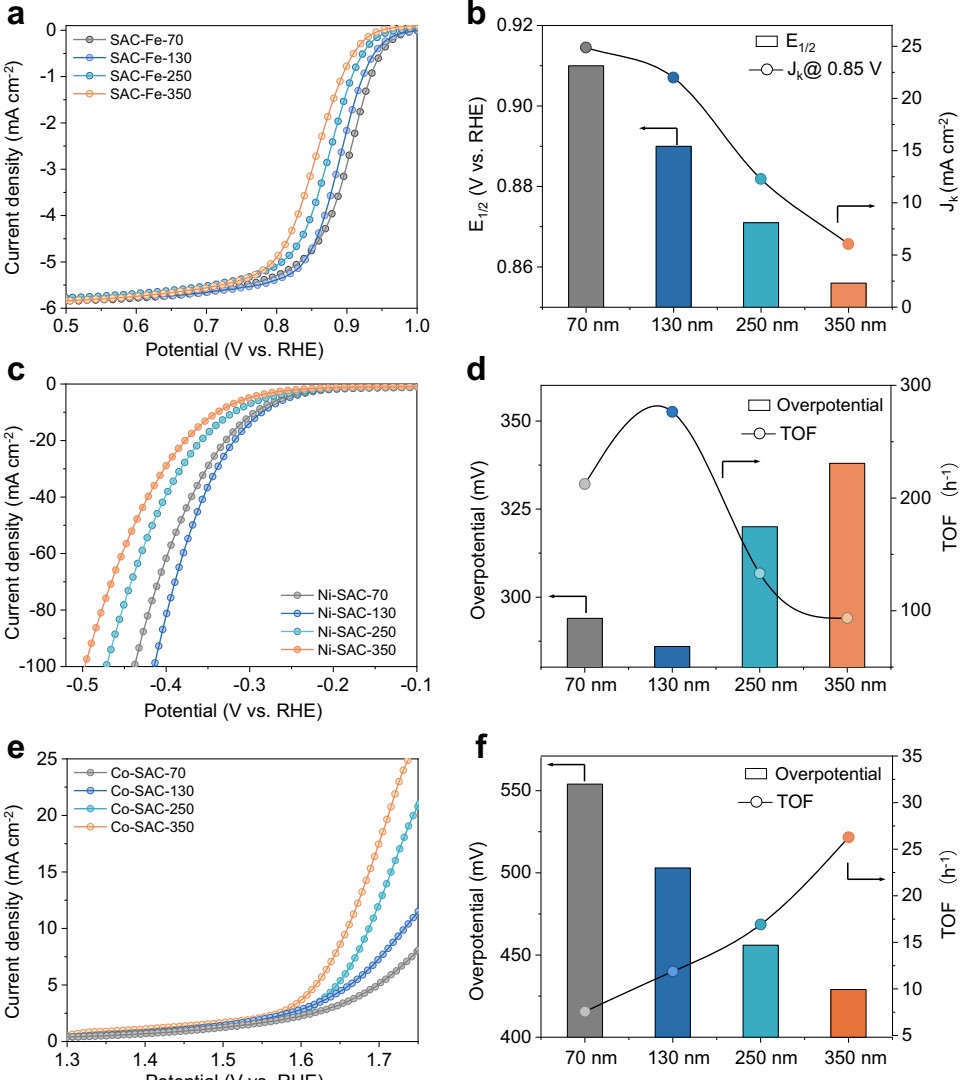

**Fig. 5 | ORR, HER and OER performance of the SACs. a** LSV curves of Fe-SACs with different sphere diameter in $O_2$-saturated 0.1 M KOH at 1600 rpm for evaluating ORR activity. **b** ORR $J_k$ at 0.85 V (vs. RHE) and $E_{1/2}$ of different Fe-SACs. **c** LSV curves of Ni-SACs with different diameter in 1 M KOH for evaluating HER activity. **d** HER overpotential at 10 mA cm$^{-2}$ and corresponding TOFs of different Ni-SACs. **e** LSV curves of Co-SACs with different diameter in 1 M KOH for evaluating OER activity. **f** OER overpotential at 10 mA cm$^{-2}$ and corresponding TOFs of different Co-SACs. Source data are provided as a Source Data file.

(727 mAh g$^{-1}$Zn). The long-term rechargeability of the Fe-SAC-70/IrC and Pt/C/IrC based batteries was tested using continuous galvanostatic discharging/charging at a current density of 5 mA cm$^{-2}$. Notably, stable performance was observed over a period of 120 h with Fe-SAC-70/IrC ZAB. This contrasts with the commercial Pt/C/IrC based ZAB system, which exhibited an obvious degradation within 40 h.

Next, we proceeded to investigate alkaline HER and OER with Ni SACs and Co SACs respectively in 1 M KOH. For alkaline HER, LSV results (Fig. 5c) show that Ni-SAC-130 displays the highest activity amongst the Ni SACs. This is summarized in Fig. 5d where Ni-SAC-130 has the lowest overpotential and highest TOF at 10 mA cm$^{-2}$. Similarly, for OER (Fig. 5e, f), we found that sphere diameter can control SAC activity, where Co-SAC-350 exhibits the lowest overpotential and highest TOF at a 10 mA cm$^{-2}$. Importantly, these results demonstrate that our nanocurvature control strategy can indeed control electrocatalyst activity for a broad range of electrocatalytic reactions.

Finally, we performed HER in acidic electrolyte (0.1 M H$_2$SO$_4$) with all 3 transition metal SAC systems (Ni/Fe/Co). In contrast to the previous reactions, we found no correlation between the sphere diameter and catalytic activity for all SAC systems (Supplementary Fig. 53). For

instance, all Ni-SACs displayed similar overpotentials for HER despite possessing different sphere diameters (nanocurvature). This is because the *H intermediate does not have a dipole moment or polarizability and hence is not affected by a change in the interfacial electric field. This is evident from our DFT calculations of the *H adsorption energy, which exhibited no significant changes as a function of interfacial electric field strength (Supplementary Fig. 54), and is consistent with prior literature[29,30]. This explains our experimental observations, where changing the sphere diameter and hence nanocurvature has no apparent effect on the acidic HER activity.

## Discussion

In this work, we first employed DFT simulations to explore the effect of interfacial electric fields on reaction pathway energetics. Using ORR and CO$_2$R as examples, we found that electric fields can control the binding energy of reaction intermediates, especially those with a high dipole moment and/or polarizability. Importantly, electric field effects operate independently of scaling relations, therefore presenting an experimental knob to control SAC activity. We reasoned that this could be implemented by hosting M−N$_4$ active sites on spherical carbon-

supports, with a smaller sphere diameter leading to a higher nano-curvature. Multiphysics simulations showed that a higher nano-curvature leads to a larger interfacial electric field. This result was experimentally confirmed through in-situ Raman spectroscopy with $SCN^-$ as the vibrational Stark shift electric field reporter under electrochemical conditions.

We then demonstrated that nanocurvature-induced electric field effects can control SAC activity for a broad range of electrocatalytic reactions including acidic $CO_2R$, alkaline ORR, alkaline HER and alkaline OER. We note that the model SAC systems developed in this work utilize only generic $M-N_4$ sites as proof of concept. Therefore, it is expected that coupling our strategy with appropriate active site design could lead to significantly higher SAC performance. Finally, we showed that nanocurvature has no significant influence on acidic HER activity, due to lack of a dipole moment and polarizability of the *H intermediate. Our findings demonstrate controllable electric field modulation as a powerful tool for controlling the activity and selectivity of SACs for a broad range of electrocatalytic reactions.

## Methods

### Materials

2,4-dihydroxybenzoic acid (≥99.9%), hexamethylenetetramine (≥99.9%), Pluronic P123 (≥99.0%), sodium oleate (≥99.0%), potassium sulfate (≥99.0%), sulfuric acid (95.0-98.0%), sodium perchlorate (≥99.0%), iron(III) acetylacetonate (≥99.9%), nickel(II) acetylacetonate (≥99.9%) and cobalt(II) acetylacetonate (≥99.9%) were purchased from Sigma-Aldrich. $Au@SiO_2$ for SHINERS experiments were purchased from XFNANO, China. Deionized water (18.2 MΩ) from an OmniaPure UltraPure Water System (Stakpure GmbH) was used for all the experiments. Carbon dioxide (99.9%), nitrogen (99.99%), argon (99.9995%) and hydrogen gas (99.9995%) were obtained from Air Liquide Singapore Pte. Ltd. Nafion 117 cation exchange membrane, and titanium screen mesh were purchased from Fuel Cell Store. The electrochemical flow cell and Ag/AgCl (3 M KCl) reference electrodes were purchased from Tianjin Aida Hengsheng Technology Development Co. The standard calibration gas mixtures for calibrating the gas chromatography system were obtained from Air Liquide Singapore Pte. Ltd. All the chemicals used in this work were of analytical grade and used without further purification.

### Characterization

Powder X-ray diffraction pattern (PXRD) was conducted on a Rigaku D/max 2500Pc X-ray powder diffractometer with monochromatized Cu Kα radiation (λ = 1.5418 Å). Scanning electron microscopy (SEM) was performed using a JEOL JSM-7610F field scanning electron microscope. Transmission electron microscopy (TEM) was obtained using a JEOL JEM-2100F field emission electron microscope working at 200 kV. Single atoms were characterized and identified with high-angle annular dark-field scanning transmission electron microscopy (HAADF-STEM) using a FEI Themis Z scanning/transmission electron microscope operated at 300 kV, equipped with a probe spherical aberration corrector. X-ray photoelectron spectroscopy (XPS) was collected on a Kratos AXIS Supra+ spectrometer equipped with a monochromatized Al Kα X-ray source and a concentric hemispherical analyzer. The survey scan was conducted using an emission current of 15 mA, pass energy of 160 eV and step size of 1 eV. The narrow scan was conducted using an emission current of 15 mA, pass energy of 20 eV and step size of 0.1 eV. The X-ray absorption spectroscopy measurements for the Co and Fe catalysts were measured at the XAFCA beamline of the Singapore Synchrotron Light Source. X-ray absorption spectroscopy measurements for the Ni catalysts were carried out at beamline 44 A of the Taiwan Photon Source (TPS), in fluorescence mode using a 7-element silicon drift detector. TOF-SIMS was performed on the TOF-SIMS 5-100S3 (ION-TOF GmbH) equipped with a reflection-type TOF analyzer. A pulsed Bi primary ion beam source at an acceleration voltage of

30 keV, 45 deg incidence, scanning area 200 μm × 200 μm. $CO_2$ reduction products were analyzed using an Agilent 8600 gas chromatography system equipped with a thermal conductivity detector and a flame ionization detector. ICP data was acquired using inductively coupled plasma-atomic emission spectroscopy (ICP-OES) on Thermo iCAP 6000.

### Synthesis of hollow polymer spheres (HPS)

The synthesis of HPS was based on a previously reported method with some modifications[51]. In a typical procedure, 2,4-dihydroxybenzoic acid and hexamethylenetetramine were dissolved in 60 ml deionized water. Supplementary Table 1 shows the required amounts of each reagent, which varies based on the intended sphere size. To this solution, 20 ml of another solution containing 30 mg Pluronic P123 and 76.4 mg sodium oleate was added under slow stirring. After slowly stirring for 10 min, the mixed solution was transferred into a 100 ml Teflon-lined stainless-steel autoclave and heated to a temperature of 160 °C for 2 h in a heating oven. After the reaction was complete, the autoclave was left to cool at room temperature. The products were then collected by centrifugation, washed three times with deionized water and ethanol and finally dried at 70 °C in a vacuum oven.

### Synthesis of single atoms on hollow nitrogen doped carbon (HNC)

Firstly, 0.2 g HPS was dispersed in 5 ml of ethanol. To this solution, 5 mL of another ethanol solution containing 7.0 mg Ni(acac)$_2$ was added under stirring. The resulting mixture was then stirred at 80 °C until all the ethanol was evaporated. After that, the resulting mixture and 4.0 g dicyandiamide were separately placed in two alumina combustion boats located at the downstream and upstream direction in a tube furnace, respectively. The tube furnace was heated to 900 °C at a heating rate of 5 °C/min under flowing nitrogen gas (10 mL/min) and held at that temperature for two hours. After cooling to room temperature, the single atom catalysts were obtained. A schematic of the synthesis process is shown in Supplementary Fig. 11.

### In-situ Raman spectroscopy experiments

Electric field measurement was carried out with a Horiba LabRam 352 Odyssey Nano Raman Spectrometer system. Measurements were performed using a custom-made in-situ cell. A classy carbon electrode coated with catalyst was used as the working electrode. A Pt wire and an Ag/AgCl were used as the counter and reference electrode, respectively. $Au@SiO_2$ was used for surface-enhanced Raman scattering for SHINERS experiments. KSCN was used as the vibrational stark effect reporter. For SHINERS, 10 μL (-0.5 mg/mL) of SHINERS was first dropped onto a glassy carbon electrode (3 mm in diameter). Following this, 5 μL of the catalyst ink (5 mg/mL) was added. After the electrode had completely dried, it was immersed in 0.1 M $NaClO_4$ containing 0.1 M KSCN for 15 min and rinsed several times with deionized water before measurement. An Olympus N2667700 water immersion objective was immersed into the electrolyte to collect the Raman spectra.

### In-situ attenuated total reflection-infrared spectroscopy (ATR-IR)

ATR-IR spectra were collected on an IRTracer-100 spectrometer (Shimadzu) equipped with an MCT detector cooled with liquid nitrogen. An Au-modified Si semi-cylindrical prism coated with catalyst was used as the working electrode and for IR reflection. The mass loading of the catalyst was 0.1 mg cm$^{-2}$ and the spectra were recorded under different working electrode potentials in $CO_2$ saturated pH 1 acidic electrolyte (0.1 M $K_2SO_4$ + $H_2SO_4$).

### Electrochemical Measurements

The $CO_2$ reduction tests were conducted in a gas diffusion electrode electrochemical flow cell system with an exposed electrode area of 1

cm$^2$. The flow cells were assembled with IrO$_x$/Ti mesh as the anode, Ag/AgCl as the reference electrode, and a Nafion exchange membrane (Nafion 117; size: 2.5 cm × 2.5 cm; thickness: 0.18 m) as the separator. The IrO$_x$/Ti mesh electrode was prepared using a dip coating and thermal decomposition procedure, according to methods described by Luc et al.[68]. The Nafion proton exchange membrane was activated in 5 wt.% H$_2$SO$_4$ at 80 °C for 2 h before use. To prepare the working electrode, 12 mg of catalyst and 12 mg carbon black were added into a mixed solution of 2 mL isopropanol/ethanol (50% (v/v)) and 200 μL Nafion solution. The resulting mixed solution was then ultrasonically treated for 2 h to form a homogeneous ink. After that, the ink was sprayed onto the GDE (Sigracet 28BC) with a mass loading of ~1 mg cm$^{-2}$ using an air brush. All electrochemical measurements were carried out using an Autolab PGSTAT204 potentiostat. During the measurement, CO$_2$ gas was passed through the cathode gas chamber at a flow rate of 100 sccm using a mass flow controller (Alicat Scientific). The FE for H$_2$ and CO is calculated based on the equation:

$$FE = \frac{n \times \upsilon \times c \times F}{i \times V_m} \quad (2)$$

where $n$ is the electron transfer number, $\upsilon$ is the gas flow rate, c is the concentration of the detected gas product, F is the Faraday constant, i is the total current and $V_m$ is the unit molar volume of gas. The outlet gas flow rate of the electrochemical cell was measured using a bubble flow meter.

The oxygen reduction test was conducted in 0.1 M KOH saturated with O$_2$ at room temperature in a three-electrode cell. A rotating disk electrode with a glassy carbon (GC) disk of 5 mm in diameter coated with catalyst was used as the working electrode. To prepare the working electrode, 5 mg of catalyst was added into a mixed solution of 1 mL isopropanol/ethanol (50% (v/v)) and 20 μL Nafion solution. The resulting mixed solution was then ultrasonically treated for 1 h to form a homogeneous ink. After that, the ink was coated on the RDE with a loading of 0.5 mg cm$^{-2}$. The cyclic voltammetry (CV) tests were first conducted with a scan rate of 50 mV s$^{-1}$. Linear sweep voltammetry (LSV) tests were tested at a rotation rate of 1600 rpm with a sweep rate of 10 mV s$^{-1}$.

The oxygen evolution and hydrogen evolution tests were conducted in 1.0 M KOH saturated with N$_2$ at room temperature in a three-electrode cell. A glassy carbon (GC) electrode of 4 mm in diameter coated with catalyst was used as the working electrode. The catalyst ink was prepared in a similar way to the ORR tests, and the ink was coated on the GC with a loading of 0.39 mg cm$^{-2}$. Before the LSV measurement, CV cycling was conducted to obtain a stable curve. All electrochemical measurements were carried out using an Autolab PGSTAT204 potentiostat.

### Zinc-air battery measurements

The Zn-air battery was assembled with Fe-SACs or 20 wt% Pt/C coated carbon paper as the air cathode, a polished Zn plate as the anode, and 6.0 M KOH aqueous solution as the electrolyte. To prepare the working electrode, 10 mg of catalyst was added into a mixed solution of 900 uL water/ethanol (50% (v/v)) and 100 μL Nafion solution. The resulting mixed solution was then ultrasonically treated for 2 h to form a homogeneous ink. After that, the ink was drop-casted onto the GDE (G60) with a mass loading of ~2 mg cm$^{-2}$. Power density and constant current discharge curves were used to characterize the performance of the Zn-air batteries. For the charge-discharge stability tests, either Fe-SAC-70 mixed with Ir/C (Fe-SAC-70: 5 wt% Ir/C = 1:1, mass ratio) or commercial catalysts (20 wt% Pt/C: 5 wt% Ir/C = 1:1, mass ratio) was used as the air cathode.

### Multiphysics simulations

The finite element numerical method to simulate interfacial electric fields was implemented under COMSOL Multiphysics. More information on these simulations can be found in the Supplementary Information (Supplementary Note 1).

### DFT calculations

DFT calculations were performed with the PBE exchange-correlation functional and the projector augmented wave (PAW) method with the Vienna ab initio simulation package (VASP)[69]. The energy cutoff of the plane wave was set to 400 eV, and 2 × 2 × 1 Monkhorst–Pack k grids were used for the Brillouin-zone integrations. The electric field (from −1.2 to 1.2 · V Å$^{-1}$) along the z-axis was considered in our calculations. The convergence criteria for the iteration process were a maximal residual force of less than 0.01 V · Å$^{-1}$ and an energy change of less than 10$^{-5}$ eV. The vacuum layer was ~18 Å. For the CO$_2$R simulation models, one layer of water molecules was added to the surface to take solvation into account. Specifically, one potassium cation and six water molecules were added near to the surface (see model structures in Supplementary Fig. 55). Additionally, the binding energy was calculated from DFT-optimized structures as follows:

$$E_{binding} = E_{CO_2^*} - (E_{slab} + E_{CO_2}) \quad (3)$$

where $E_{CO_2^*}$ is the energy of the system with CO$_2$ proximate to the slab surface, $E_{slab}$ is the energy of the M-N$_4$ surface (with and without K$^+$ and six water molecules for the respective cases), and $E_{CO_2}$ is the gas-phase energy of CO$_2$. More information can be found in the Supplementary Information (Supplementary Note 3).

## Data availability

The authors declare that the data supporting the findings of this study are available within the paper and its Supplementary Information files. Should any raw data files be needed in another format they are available upon request. Source data are provided with this paper.

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

## Acknowledgements

Y.L. and J.Z. acknowledge support and funding from the A*STAR (Agency for Science, Technology, and Research) under its LCERFI program (Award No: U2102d2002). Y.L. acknowledges support and funding from the NRF Fellowship (Award No: NRF-NRFF14-2022-0003). We acknowledge the use of the XAFCA beamline of the Singapore Synchrotron Light Source (SSLS) for the collection of the XAS data used in this work. We acknowledge the use of high-performance computational facilities from the National Supercomputing Centre (NSCC) Singapore. We thank Lei Chen and Lei Wang for their assistance with the in-situ ATR-IR measurements. We thank Yuanjie Pang for their assistance with the finite element numerical method model.

## Author contributions

Y.L. supervised the project. Y.L. and B.W. conceived the idea and designed the experiments. B.W. performed all the experimental work. M.W. and Jia Z. carried out and supervised the DFT simulations respectively. Z.F. performed the Multiphysics simulations. C.M. performed the AC-TEM characterization. S.X. and L-Y.C. carried out the XAS experiments. M.Z. performed the XPS measurements. N.L. assisted in Raman spectroscopy experiments. Z.M. and S.C. assisted in catalyst preparation. W.R.L and D.W. contributed to data analysis and manuscript editing. Y.L and B.W. co-wrote the manuscript. All authors discussed the results and assisted during the manuscript preparation.

## Competing interests

The authors declare no competing interests.
