## [Peer Review File · Nature Communications]

Nanocurvature-induced field effects enable control over the activity of single-atom electrocatalystsREVIEWER COMMENTS

Reviewer #1 (Remarks to the Author):

The authors of this work used multiple methods to characterize, test and rationalize the performance of magnetized single-atom catalysts (namely, Ni, Fe, and Co) supported on spherical carbon supports for different electrochemical reactions such as CO₂RR, ORR, OER and HER. The strongest claim here by the authors is this work clearly shows how the curvature of these carbon spheres can be used to modulate the electric field effects of these SACs for targeted reactions that involve polarizable intermediates.

While this work is definitely interesting, I do have some deep concerns for a few technical issues listed here:

1. From the beginning, the authors claim how DFT can/may help to explain the effect of interfacial electric fields. I am quite sure that this has been studied a lot in literature (as the authors themselves have cited quite a few of the known ones). They then claim that the results for the SAC systems mentioned in this paper may be translated to the scaling plots to justify their next experiments. Here I would like to point out a recent paper on such field effects (DOI 10.1021/acscatal.2c00997) where one has to go beyond the standard considerations where I quote "charge rearrangement upon specific adsorption also affects the local field experienced by the adsorbate, which precludes estimates merely based on the local atomic-scale electrode geometry". This is the question for the authors: How is this important for your work and how this can change your interpretation of your results. From my point of view, I think it is quite native from the DFT+FEM type calculations to draw conclusions in this work to explain the physical origin of the mechanisms in their experiments. I am not convinced that the DFT+FEM results presented here can be suggestive (or even conclusive) for what the authors have claimed.
2. In relation to the question above, it is also not discussed (at least from their theoretical work) how stable are these carbon supported catalysts "operando" under a field. There have been suggestions that detaching of the SACs were reported/suggested and from their DFT calculations, this is not discussed or reported. I do understand that it is mentioned in the manuscript of how a stable performance of these catalysts is achieved for long hours under technical conditions. However, it does not provide new insights as to how this may be possible in contrast to the Pt/C system.
3. From the HAADF-STEM plots, it is also not 100% clear if the metal is clearly of a single-atom nature. I admit and agree that one may not expect a "bulk-like clustering" of metal particles but it is may be "double-atoms"? "Triple-atoms"? How can one experimentally demonstrate this clearly?
4. There is also growing evidence that supported SACs (especially under operando electrochemical environment) are often ligated with small molecules of solvents and may have an impact on the chemistry of the SAC on the reactions studied in this work. I wonder if the authors could comment on how such reaction environment will have on the performance of the SAC for reactions such as CO₂RR, ORR, OER and HER.
5. From the choice of Ni, Fe and Co, it is also not clearly mentioned how the magnetic spin states (if any) on these magnetic SACs helps/assists/or not the catalysis and how their spin state in these SAC may be influenced by the curvature and applied field.
6. Lastly, how the authors so "causally" mentioned that these curvature/field engineered SACs work well for all reactions except HER in just a sentence is quite "surprising". For a paper in a journal of such high profile, a detailed discussion or explanation must and should be provided as to why and how this is the case. More theoretical/experimental evidence from the authors are required.

Reviewer #2 (Remarks to the Author):

The numerical modeling performed in this work suggests that higher curvature of spherical electrocatalysts uniformly enhances the electric field at the electrode-electrolyte interface. Based on the DFT simulations, this modulation of the interfacial electric field alters the adsorption energy of intermediates in CO₂ and O₂ reduction reactions differently. The authors suggest that the activity and selectivity toward a specific electrochemical reaction can be optimized by tuning the diameter of spherical electrocatalysts. These claims were also supported experimentally using SACs with Ni, Fe, and Co as the active material and spherical carbon with varying size distributions as the support. The dependence of interfacial electric field to curvature was experimentally investigated using Kelvin probe atomic force microscopy and in-situ Raman spectroscopy. Finally, the performance of the catalyst materials was studied in CO₂R to CO, ORR, OER, and HER. Based on electrochemical results, the authors concluded that changing the curvature of their SACs successfully adjusts the performance toward various electrochemical reactions if involved intermediates possess dipole moments and polarizability to interact with the interfacial electric field. This paper is suitable for Nature Communications, however the following concerns should be addressed prior to publication.

1. Based on the conditions used in this work, the mixed effects of counterions and the local chemical environment should be considered and discussed in the text [see e.g., Nature Catalysis 1.12 (2018): 946-951; Nature Catalysis 2.8 (2019): 702-708; Nature Catalysis 5.4 (2022): 268-276]. Spheres with smaller radii pack differently compared to larger spheres, creating different diffusion rates and microenvironments. Therefore, the chemical environment is not the same around catalysts with different curvatures, which makes the electrochemical data convoluted and not unambiguously supportive of the claims of the paper. In other words, the experimental data supports the fact that the diameter of spherical particles affects the electrochemical performance, but the results are not justifiable solely based on the effect of the interfacial electric field on the energy of adsorbates to the exclusion of other factors at play, as they have not been ruled out.
2. When performing Finite Element Analysis, mesh sensitivity analysis must be performed to ensure the convergence of numerical results. The final mesh size used for the simulations should be reported as well. However, since particles studied in this work have spherical symmetry, it is suggested that the authors use Poisson-Nernst-Planck equations in spherical or polar form and derive an analytical solution for electric field distribution as a function of particle radius.
3. Based on the DFT simulations, the adsorption energy of intermediates is modulated by the electric field. These intermediates are located within the Stern layer. However, FEA simulations are performed for the diffuse part of the electrical double layer. The authors need to redo their electric field simulations considering the Stern layer or provide the reasoning for not considering the Stern layer.
4. Experimental data in Figure 4a-d does not clearly support the DFT simulations and the effect of the curvature-enhanced electric field. Based on DFT simulation, stronger electric fields reduce the energy barrier of the rate-determining step for CO₂R. It should result in higher activity or Turnover Frequency (TOF) for catalysts with higher curvature. However, an intermediate curvature shows the highest activity in all catalysts, suggesting that the interpretation of experimental results is not possible solely based on the curvature-modulated electric field effect. In lines 273-276, the authors assert "Such a volcanic relationship is expected since there should exist an optimal value of the interfacial electric field where reaction energy barriers between all the reaction intermediates are the lowest." However, there is no data supporting this explanation.
5. The claim that the energy of adsorbed H* is not affected by an interfacial electric field should be supported by DFT calculations.
6. The SHINERS technique used for measuring the interfacial electric field is novel and not well-established, according to the authors. It should be clearly explained in the text why this technique was chosen, and how SiO₂-coated gold particles were applied to the samples. It is also suggested to include the spectrum of the catalyst without adsorbed SCN⁻ as a control. Error bars should be added to the data in Figure 3d.
7. The interfacial electric field increases the concentration of counter-ions in the vicinity of the

electrode [Nature 537, no. 7620 (2016): 382-386]. These counter-ions can alter the adsorption energy of intermediates, which was not considered in the DFT simulations. This should be discussed in the text.

8. In Supplementary Note 1, the authors have mentioned the Electric Current Module for their simulation; it should be explained how this module was used for the electric field simulation that was performed, as the relevance is unclear.

9. Statistical analysis of the nanoparticle size must be performed until the Gaussian shape of particle size distribution is achieved.

Reviewer #3 (Remarks to the Author):

In this manuscript, the authors reported the preparation of single atom catalysts (SACs) consisting of M-N4 sites anchored on a series of spherical hollow carbon supports with different diameters, and studied the effect of nanocurvature of carbon supports on electrocatalytic activities of SACs (CO₂RR, OER, and ORR). They try to attribute the activity modulation of SACs to nanocurvature induced interfacial electric field via theoretical calculations. However, there is a large discrepancy between the DFT models (flat surface) and the actual structure (curved surface) of the catalysts. The conclusions drawn from DFT calculations are contradictory with those of experimental results, in which the minimum curvature of carbon support corresponds to best performances of SACs in DFT calculations, while a volcanic performance trend is experimentally obtained. As a result, I am not convinced by the mechanism. Some extra comments are listed below:

1. There are already several related reports concerning the curvature of supports modulated catalytic activities of SACs, such as Nat. Commun. 12, 6335 (2021), Angew. Chem. Int. Ed. 2021, 60, 22722–22728, Nat. Catal. (2023). <https://doi.org/10.1038/s41929-023-01005-3>, and ACS Energy Lett. 2023, 8, 1330–1335, which largely weaken the novelty of the current work.

2. It is important to elucidate the intermediate species and the mechanism of the interfacial electric field on the intermediate species, and thus in-situ Fourier transform infrared spectroscopy is necessary.

3. What are the thicknesses of spherical carbon supports and are they consistent? If there is a difference in thickness, what is the effect of thickness on catalytic performance?

4. For better contrast, the CO₂R performance of Fe-SACs and Co-SACs with diameters of 70, 130, 250 and 350 nm should also be provided.

5. Why is the DFT study of CO₂R focused on the Ni-N4 SACs?

6. Considering the differences in dimensions of different samples, the authors should standardize the scale in KPFM. Furthermore, there are some problems in the unit and legends, such as Fig. 3d and Fig. 5a.

7. In Fig. 1a, the relative position of Gibbs free energy in the initial and final state is not correct. Please check them.

8. Some minor errors:

Fig. 1d, line 50, VÅ⁻¹ instead of eVÅ⁻¹.

Line 458, Table S3 instead of Table S1.

We thank the editor for handling our manuscript and the reviewers for their comments, which we have used to improve the quality of the work. Changes within the manuscript and supporting information are highlighted in yellow. Below is a point by point response to the reviewer comments, which are written in blue font.

Reviewer #1 (Remarks to the Author):

The authors of this work used multiple methods to characterize, test and rationalize the performance of magnetized single-atom catalysts (namely, Ni, Fe, and Co) supported on spherical carbon supports for different electrochemical reactions such as CO₂RR, ORR, OER and HER. The strongest claim here by the authors is this work clearly shows how the curvature of these carbon spheres can be used to modulate the electric field effects of these SACs for targeted reactions that involve polarizable intermediates.

While this work is definitely interesting, I do have some deep concerns for a few technical issues listed here:

Response

We are grateful to Reviewer #1 for their positive evaluation of our manuscript and interesting suggestions.

1. From the beginning, the authors claim how DFT can/may help to explain the effect of interfacial electric fields. I am quite sure that this has been studied a lot in literature (as the authors themselves have cited quite a few of the known ones). They then claim that the results for the SAC systems mentioned in this paper may be translated to the scaling plots to justify their next experiments. Here I would like to point out a recent paper on such field effects (DOI 10.1021/acscatal.2c00997) where one has to go beyond the standard considerations where I quote "charge rearrangement upon specific adsorption also affects the local field experienced by the adsorbate, which precludes estimates merely based on the local atomic-scale electrode geometry". This is the question for the authors: How is this important for your work and how this can change your interpretation of your results. From my point of view, I think it is quite native from the DFT+FEM type calculations to draw conclusions in this work to explain the physical origin of the mechanisms in their experiments. I am not convinced that the DFT+FEM results presented here can be suggestive (or even conclusive) for what the authors have claimed.

Response

We thank the reviewer for bringing to our attention the relevant and important paper by Beinlich et al., which we have read with great interest and now cite in our manuscript. In this theoretical study, the authors have 5 main findings:

(1) The adsorption energies of intermediates change with applied potential and this can be treated as an effective dipole-field interaction.

(2) Due to charge rearrangement, the surface dipole that forms upon adsorption (rather than the intrinsic molecular dipole moment of the adsorbate) is the true value that needs to be considered when examining these dipole-field interactions.

(3) At the same time, charge rearrangement also means that the net charging of the adsorption site becomes altered upon adsorption as well, and this is very important when evaluating and comparing atomic scale protrusions (e.g. kink) to that of flat adsorption sites (e.g. terrace).

(4) Simple estimates of the potential dependence of the adsorption energy can be made based on the adsorbate-induced work function change at the point of zero charge.

(5) Much of the current literature examines reaction mechanisms without considering potential dependent adsorption energy changes as a result of dipole-field interactions. The authors point out that these can reach up to 300 m eV/V, which is certainly non-trivial and hence could have a significant impact on reaction pathways.

We find that (1) and (5) is consistent with our work, where we show that electric fields can indeed influence adsorption energies and needs to be considered when evaluating reaction mechanisms. Similarly, in our work, intermediates considered have dipole moments, which interact with the electric field and alters their adsorption energies.

We also find that (2) and (4) is consistent with our work. Beinlich et al. show that the surface dipole of CO changes from the intrinsic molecular dipole moment of 0.1 to estimated values of -0.29, -0.16 and 0.02 for terrace, step and kink on Pt respectively. Similarly, we find in our case of *CO on Ni-N₄ catalyst, we find that the fitted dipole moment is -0.052, rather than the intrinsic molecular dipole moment of 0.1.

While (3) is interesting and highly relevant to metallic systems, we find that this does not quite apply to our work. This is because our adsorption sites (catalytic sites) for our Ni-SAC systems are quite uniform and should all consist of Ni-N₄. As a result, we expect that the adsorption energies, charge rearrangement, net charging of the adsorption site and effective dipole moment of intermediates should be quite similar across all the catalytic sites of our catalyst system.

Taken together, we find that our main conclusions are consistent with the work by Beinlich et al, especially with regards to the notion that the applied potential can change intermediate adsorption energies, which can in turn significantly impact reaction mechanisms.

The related reference has been cited in the revised manuscript and listed below:

30 *Beinlich, S. D., Hörmann, N. G. & Reuter, K. Field effects at protruding defect sites in electrocatalysis at metal electrodes? ACS Catal. 12, 6143-6148 (2022).*

2. In relation to the question above, it is also not discussed (at least from their theoretical work) how stable are these carbon supported catalysts "operando" under a field. There have been suggestions that detaching of the SACs were reported/suggested and from their DFT calculations, this is not discussed or reported. I do understand that it is mentioned in the manuscript of how a stable performance of these catalysts is achieved for long hours under technical conditions. However, it does not provide new insights as to how this may be possible in contrast to the Pt/C system.

Response

The deactivation of single-atom catalysts via demetallation is certainly an issue of importance, especially when considering electrolyzer operation over extended durations. For instance, Holby et al. employed a computational approach to study the stability of M-N-C active sites hosted on a graphene support for the oxygen reduction reaction (*ACS Catal.* 2020, 10, 14527).

Here, we took a similar approach to study the stability of Ni-N₄ active sites under an electric field. Specifically, we investigated the total energy change of Ni-N₄ as a function of interfacial electric field strength. Our results are plotted in Supplementary Fig. 46 (reproduced below), where we observed that the total energy of Ni-N₄ became more negative with increasing interfacial electric field strength. Hence, this result indicates that Ni-N₄ is indeed stable in the presence of an electric field.

Supplementary Fig. 46 | The normalized total energy of the Ni-N₄ active site as a function of interfacial electric field strength.

Furthermore, our post-experiment characterization of our Ni-N₄ catalyst was also carried out with transmission electron microscopy (TEM), aberration corrected high-angle annular dark-field (AC-HAADF) microscopy and X-ray absorption spectroscopy (XAS) as shown in Supplementary Fig. 45 (reproduced below). From these results, we observed no significant changes in the structure, morphology and Ni-N₄ active sites before and after CO₂R. In the EXAFS spectra (Supplementary

Fig. 45d), only a main peak belonging to atomically dispersed Ni on the support was observed, with no peaks related to Ni-Ni that would be observed if Ni nanoparticles or nanoclusters were present. Hence, the catalyst is stable and remains unchanged after the reaction. The Ni is still atomically dispersed on the support rather than transformed into nanoparticle or nanoclusters. For comparison, we have also reproduced below the XAS data of the pristine catalyst before reaction (Fig. 2h-2i).

Supplementary Fig. 45 | Post stability test characterization of Ni-SAC-250. (a) TEM image, (b) AC-HAADF-STEM image, (c) Ni K-edge XANES spectra and (d) Ni K-edge Fourier-transformed (FT) k^2 -weighted $\chi(k)$ functions.

Fig. 2h and 2i | (h) Ni K-edge XANES spectra and (i) Ni K-edge Fourier-transformed (FT) k^2 -weighted $\chi(k)$ functions of Ni-SACs with different sphere diameters. NiO and Ni foil were used as references.

3. From the HAADF-STEM plots, it is also not 100% clear if the metal is clearly of a single-atom nature. I admit and agree that one may not expect a "bulk-like clustering" of metal particles but it is may be "double-atoms"? "Triple-atoms"? How can one experimentally demonstrate this clearly?

Response

Since the HAADF-STEM image is a 2D projection view instead of a 3D view, it is hard to say for certain whether each white spot corresponds to a single atom and this is a subject of much research efforts in the field. Therefore, besides HAADF-STEM imaging, X-ray absorption spectroscopy is required to characterize the coordination of the metal atoms present in the sample. This is required because it is used to verify that the catalyst indeed consists of only single atoms, without any peaks that could be correlated to metal-metal bonding.

Hence, whether or not the atoms present exist as double-atoms or triple-atoms or not is difficult to say for certain based on the HAADF-STEM images alone. However, as suggested by previous work (*J. Am. Chem. Soc.*, 2017, 139, 17281; *J. Am. Chem. Soc.* 2022, 144, 40, 18485), if double-atoms or triple-atoms are present in the sample, a metal-metal interaction path belonging to double or triple-atoms should be observed from the EXAFS spectrum. However, we note the absence of a peak belonging to such metal-metal interactions in our EXAFS results (Figure 2h, i), and this therefore indicates that the metal is dispersed as single-atoms in the catalyst.

We have added the above discussion into the revised manuscript, which is also listed below:

We note the absence of a peak belonging to the second shell Ni-Ni interaction, and this confirms that Ni is dispersed as single-atoms rather than clusters, dual-atoms or triple-atoms in the SACs.

4. There is also growing evidence that supported SACs (especially under operando electrochemical environment) are often ligated with small molecules of solvents and may have an impact on the chemistry of the SAC on the reactions studied in this work. I wonder if the authors could comment

on how such reaction environment will have on the performance of the SAC for reactions such as CO₂RR, ORR, OER and HER.

Response

There has been a recent theoretical report by Misra et al., which suggests that besides solvating intermediates, solvent water can also act as a ligand that interacts directly and competes with the CO₂ molecule to bind to the single metal atom active site (*J. Catal.* 2023, 422, 1). The authors investigated this on a series of transition metal (M) atoms supported on a nitrogen doped graphene structure, with M-N₄ type active sites. They found that if the interaction between M and H₂O is strong, H₂O can compete with CO₂ or other reaction intermediates for binding with the single metal atom active site. Specifically, this results in less negative free energies of adsorption for CO₂, COOH and OCHO intermediates, which in turn has a significant impact on energy barriers in the reaction pathway. For instance, the relative stability of the competing COOH and OCHO intermediates can change depending on whether H₂O was considered or not.

Such a scenario could potentially be relevant in our work, where we performed all our electrocatalytic reactions exclusively in aqueous media and our catalysts also consist of M-N₄ active sites. While the effect of solvent water directly binding to our transition metal atom sites could also be evaluated in our theoretical work by following a similar approach adopted by Misra et al., we find that this requires an extensive and considerable amount work, which we feel is out of scope for the purpose and goal of this work. In addition, we note that the adsorption energy of solvent water should also be affected by interfacial electric fields since solvent water has a dipole moment, making it also susceptible to dipole-field interactions. This will hence affect the competition between intermediate binding and H₂O binding to the transition metal single atom site, which was not previously considered by Misra et al. and will need to be evaluated in our case. Finally, we also believe that considering this solvent binding competition would not alter the primary conclusion of our work, which is that the nanocurvature of the carbon support can influence interfacial electric fields, which will in turn alter the adsorption energies of reaction intermediates and affect catalytic activity.

5. From the choice of Ni, Fe and Co, it is also not clearly mentioned how the magnetic spin states (if any) on these magnetic SACs helps/assists/or not the catalysis and how their spin state in these SAC may be influenced by the curvature and applied field.

Response

This is an interesting question and there are indeed some works that have studied the effect of the spin states of magnetic single-atom catalysts on their activity. For example, Zhai et al. reported the spin state of Fe-NC can be tuned by S doping. Compared to the original Fe-NC, the Fe in the S-doped catalyst has a lower spin state, which benefits desorption of the *OH intermediate, thus enhancing the ORR activity of the resulted catalyst (*Angew. Chem. Int. Ed.* 2021, 133, 25608). Recently, Liu et al. also reported tuning of the spin of Co in a binuclear cobalt phthalocyanine

catalyst. They found that the higher spin binuclear cobalt phthalocyanine catalyst shows a better methanol selectivity (*Nat Commun* 2023,14, 6550). Notably, spin tuning was observed to be accompanied by changes in the electronic structure change of the catalyst, which can be evaluated using characterization methods such as X-ray absorption spectroscopy. Hence on this basis, we find that spin tuning does not happen in our work. In our work, we regulated the curvature of the carbon support, which does not alter the electronic structure of the metal site. This is evident from our EXAFS and XPS results (Figure 2h, i and Supplementary Fig. 18d), where the edge position or the binding energy of the metal sites has no apparent effect on the catalysts of different nanocurvature. Therefore, we believe that altering the nanocurvature of the carbon sphere would not affect the spin state of the metal sites.

We have added the above discussion into the revised manuscript, which is also listed below:

Furthermore, since the XAS data reveals that there are no major differences in the electronic structure of all four Ni-SAC samples, we infer that their magnetic spin states should be similar as well.

6. Lastly, how the authors so "causally" mentioned that these curvature/field engineered SACs work well for all reactions except HER in just a sentence is quite "surprising". For a paper in a journal of such high profile, a detailed discussion or explanation must and should be provided as to why and how this is the case. More theoretical/experimental evidence from the authors are required.

Response

In our experiments, we found that the nanocurvature of our catalysts had a strong impact on the HER activity under alkaline conditions (Fig. 5c). HER in alkaline media involves the Volmer step going through H₂O, which has a dipole moment. Hence, we postulate that for this reason, we observe that the nanocurvature has a strong impact on the alkaline HER activity.

On the other hand, we do not observe a significant impact of nanocurvature on the HER activity under acidic conditions. In this case, H⁺ are the active species being reduced rather than H₂O, thus the relevant intermediate is *H. We reasoned that this is because *H has no dipole moment or charge, hence an electrostatic interaction would not have a strong influence on the *H adsorption energy and hence the acidic HER activity.

To further investigate this, we now provide new DFT calculations on the *H adsorption energy on Ni-N₄, Fe-N₄ and Co-N₄ as a function of interfacial electric field strength (Supplementary Fig. 49, reproduced below). We observe that the adsorption energy of *H on Ni-N₄, Co-N₄ and Fe-N₄ does not change significantly with increasing electric field strength. Our results are consistent with a previous computational result (*J. Am. Chem. Soc.* 2017, 139, 32, 11277). Importantly, this helps to explain why the acidic HER activity is not significantly affected by the catalyst nanocurvature.

We have added the above results and discussion into the revised manuscript, which are also listed below:

This is because the *H intermediate does not have a dipole moment or polarizability and hence is not affected by a change in the interfacial electric field. This is evident from our DFT calculations of the *H adsorption energy, which exhibited no significant changes as a function of interfacial electric field strength (Supplementary Fig. 49), and is consistent with prior literature^{29, 65}.

Supplementary Fig. 49 | *H adsorption energy as a function of electric field strength on Ni-N₄, Fe-N₄, and Co-N₄.

Reviewer #2 (Remarks to the Author):

The numerical modeling performed in this work suggests that higher curvature of spherical electrocatalysts uniformly enhances the electric field at the electrode-electrolyte interface. Based on the DFT simulations, this modulation of the interfacial electric field alters the adsorption energy of intermediates in CO₂ and O₂ reduction reactions differently. The authors suggest that the activity and selectivity toward a specific electrochemical reaction can be optimized by tuning the diameter of spherical electrocatalysts. These claims were also supported experimentally using SACs with Ni, Fe, and Co as the active material and spherical carbon with varying size distributions as the support. The dependence of interfacial electric field to curvature was experimentally investigated using Kelvin probe atomic force microscopy and in-situ Raman spectroscopy. Finally, the performance of the catalyst materials was studied in CO₂R to CO, ORR, OER, and HER. Based on electrochemical results, the authors concluded that changing the curvature of their SACs successfully adjusts the performance toward various electrochemical reactions if involved intermediates possess dipole moments and polarizability to interact with the interfacial electric field. This paper is suitable for Nature Communications, however the following concerns should be addressed prior to publication.

Response

We are thankful to Reviewer #2 for their positive evaluation of our manuscript and thoughtful suggestions.

1. Based on the conditions used in this work, the mixed effects of counterions and the local chemical environment should be considered and discussed in the text [see e.g., Nature Catalysis 1.12 (2018): 946-951; Nature Catalysis 2.8 (2019): 702-708; Nature Catalysis 5.4 (2022): 268-276]. Spheres with smaller radii pack differently compared to larger spheres, creating different diffusion rates and microenvironments. Therefore, the chemical environment is not the same around catalysts with different curvatures, which makes the electrochemical data convoluted and not unambiguously supportive of the claims of the paper. In other words, the experimental data supports the fact that the diameter of spherical particles affects the electrochemical performance, but the results are not justifiable solely based on the effect of the interfacial electric field on the energy of adsorbates to the exclusion of other factors at play, as they have not been ruled out.

Response

Firstly, to investigate the effect of the local chemical environment, we prepared a series of electrodes with different catalyst loadings. In each case, since the catalyst used is identical but the loading is different, the only condition that changes is the local chemical environment. Specifically, we selected 3 different loadings of 0.2, 0.5 and 1.0 mg cm⁻² for the four catalysts: Ni-SAC-70, Ni-SAC-130, Ni-SAC-250 and Ni-SAC-350. CO₂R was then performed with these electrodes. We found that even when the loading was changed, the trend of the FE for the different diameter Ni single-atom catalysts remained the same (Supplementary Fig. 42, reproduced below). Specifically, we observed that Ni-SAC-250 exhibited the highest FE to CO regardless of the loading used. This

suggests that the local chemical environment does not play the main dominant role in influencing the CO₂R performance for our single-atom catalysts.

To investigate the effect of counterions, we performed CO₂R experiments with different concentrations of K₂SO₄ added to the electrolyte. As shown in Supplementary Fig. 42 below, the Ni-SAC-250 catalyst still shows the highest FE to CO in these different concentrations of K₂SO₄. This indicates that the counterions are also not the main factor in dictating the CO₂R performance of our single-atom catalysts.

Based on these results, we show that the catalyst sphere diameter plays a more dominant role in influencing the CO₂R performance.

We have added the above results and discussion into the revised manuscript, which are also listed below:

The local chemical environment and presence of counterions have been previously indicated as factors that affect the CO₂R performance^{20,64,65}. To investigate the extent to which these factors influence the activity of our Ni-SAC catalysts, two control experiments were conducted. Firstly, to study the role of the local chemical environment, we prepared electrodes with different catalyst loadings of 0.2, 0.5 and 1.0 mg cm⁻² for each of the Ni-SACs. Secondly, to study the role of counterions, we tested the Ni-SAC catalysts with different concentrations of K₂SO₄ in the electrolyte. From the results (Supplementary Fig. 42), we found that regardless of the catalyst loading and counterion concentration, Ni-SAC-250 exhibits the highest FE to CO. Hence, these results indicate that the catalyst sphere diameter plays the more dominant role in influencing the CO₂R performance.

The related references have been cited in the revised manuscript and listed below:

- 20 Gu, J. et al. Modulating electric field distribution by alkali cations for CO₂ electroreduction in strongly acidic medium. *Nat. Catal.* **5**, 268-276 (2022).
- 64 Zhuang, T.-T. et al. Copper nanocavities confine intermediates for efficient electrosynthesis of C₃ alcohol fuels from carbon monoxide. *Nat. Catal.* **1**, 946-951 (2018).
- 65 Wang, L. et al. Electrochemically converting carbon monoxide to liquid fuels by directing selectivity with electrode surface area. *Nat. Catal.* **2**, 702-708 (2019).

Supplementary Fig. 42 | FE towards CO for Ni-SACs with different sphere diameters under various conditions. (a-c) Different catalyst loadings and (d-f) different concentrations of K_2SO_4 in the electrolyte (pH=1 H_2SO_4 solution) at -1.5 V vs RHE.

2. When performing Finite Element Analysis, mesh sensitivity analysis must be performed to ensure the convergence of numerical results. The final mesh size used for the simulations should be reported as well. However, since particles studied in this work have spherical symmetry, it is suggested that the authors use Poisson-Nernst-Planck equations in spherical or polar form and derive an analytical solution for electric field distribution as a function of particle radius.

Response

In our simulation, triangular meshes were applied with a maximum mesh size of 200 nm, a minimum size of 0.1 nm and a maximum growth rate of 1.3. Hence, for a sphere with a radius of 50 nm, the mesh number is 3318. We also added the mesh sensitivity analysis by running the same simulations with different mesh size (see Supplementary Fig. S50 and Supplementary Table S21 in Supplementary Note 1). With a smaller maximum mesh size and maximum element growth rate, the number of elements increased. However, we found that this did not influence the simulation results, indicating that the mesh created was good enough to ensure convergence of numerical results. We have added these results into the Supplementary Note 1, which are reproduced below:

Supplementary Fig. 50 | The electric field distribution vs sphere radius under different mesh sizes.

Supplementary Table 21. Parameters of each different mesh.

Mesh	Maximum mesh size	Minimum mesh size	Maximum growth rate
1	200	0.08	1.1
2	200	0.1	1.3
3	400	1	1.3
4	800	10	1.3

It is a good suggestion to use the Poisson-Nernst-Planck equations to derive an analytical solution for the electric field distribution as a function of particle radius for the spherical symmetry structure. However, we feel that this is not suitable in our case because we need to consider the electrolyte when modeling the electric field distribution. Specifically, we included 0.1 M K₂SO₄ and 0.05 M H₂SO₄ as the electrolyte in our model, which cannot be easily considered and included in the Poisson-Nernst-Planck equations.

3. Based on the DFT simulations, the adsorption energy of intermediates is modulated by the electric field. These intermediates are located within the Stern layer. However, FEA simulations are performed for the diffuse part of the electrical double layer. The authors need to redo their electric field simulations considering the Stern layer or provide the reasoning for not considering the Stern layer.

Response

We thank the reviewer for bringing up this point, as our description may have been somewhat unclear. We had actually already considered the Stern layer in our simulation. In the Gouy-Chapman-Stern model of the electric double layer, the thickness of the Stern layer (Helmholtz layer) was identical to the radius of a hydrated cation. Based on the Poisson-Nernst-Planck equations, in the Helmholtz layer,

$$\nabla^2 V = 0$$

And in the diffuse layer,

$$\nabla^2 V = \rho_V = F(c_+ - c_-)$$

Detailed information on the model is included in Supplementary Note 1.

4. Experimental data in Figure 4a-d does not clearly support the DFT simulations and the effect of the curvature-enhanced electric field. Based on DFT simulation, stronger electric fields reduce the energy barrier of the rate-determining step for CO₂R. It should result in higher activity or Turnover Frequency (TOF) for catalysts with higher curvature. However, an intermediate curvature shows the highest activity in all catalysts, suggesting that the interpretation of experimental results is not possible solely based on the curvature-modulated electric field effect. In lines 273-276, the authors assert “Such a volcanic relationship is expected since there should exist an optimal value of the interfacial electric field where reaction energy barriers between all the reaction intermediates are the lowest.” However, there is no data supporting this explanation.

Response

Based on point 7 of the reviewer comments, a question was raised regarding the effect of counterions on CO₂ reduction. For instance, work by Sargent et al. [Nature 537, no. 7620 (2016): 382-386] showed that the presence of K⁺ can result in higher CO₂ reduction activity and selectivity. Hence in response to point 4 and point 7, we revised our DFT calculations by including K⁺ and H₂O into the simulation model.

As shown in Fig. 1a (reproduced below), among the four elementary reaction steps, the formation of *COOH is calculated to be the rate-determining step (RDS). The free energy change in this step decreased from 0.96 eV to 0.88 eV when the electric field increased from 0 to 0.4 VÅ⁻¹. However, when the electric field was increased to -0.8 VÅ⁻¹, the free energy change in this step increased to 0.9 eV. This indicates there is an optimal electric field strength for CO₂R performance.

In addition, the limiting potential difference between CO₂R and HER (U_L(CO₂)-U_L(H₂); U_L= -ΔG₀/e) has also been calculated and employed as the descriptor of CO selectivity according to prior literature (*Angew. Chem. Int. Ed.* 2019, 58, 6972; *J. Am. Chem. Soc.* 2017, 139, 24, 8329), where a more positive value of U_L(CO₂)-U_L(H₂) represents a higher CO₂R selectivity. As shown in Fig. 1b, an optimal electric field strength was observed for this descriptor with Ni-N₄ and is qualitatively consistent with our experimental results.

We have added the above results and discussion into the revised manuscript, which are also listed below:

For example, the free energy change for this step (limiting potential) is reduced from 0.96 eV to 0.88 eV when an electric field of -0.4 VÅ⁻¹ is applied, but increased to 0.9 eV when the electric field was further increased to -0.8 VÅ⁻¹. In addition, the limiting potential difference between CO₂R and HER (U_L(CO₂)-U_L(H₂); U_L = -ΔG₀/e) was also calculated and employed as the descriptor for CO selectivity^{45,46}, with a more positive value of U_L(CO₂)-U_L(H₂) indicating a higher CO₂R

selectivity. These findings are summarized in Fig. 1b, where an optimal electric field strength was observed for this descriptor with Ni-N₄.

The related references have been cited in the revised manuscript and are listed below:

- 45 Kim, D. et al. Electrochemical activation of CO₂ through atomic ordering transformations of AuCu nanoparticles. *J. Am. Chem. Soc.* **139**, 8329-8336 (2017).
- 46 Ren, W. et al. Isolated diatomic Ni-Fe metal–nitrogen sites for synergistic electroreduction of CO₂. *Angew. Chem. Int. Ed.* **58**, 6972-6976 (2019).

Fig. 1 | (a) Reaction pathway Gibbs free energy diagrams and (b) theoretical limiting potential difference of U_L(CO₂)-U_L(H₂) on Ni-N₄ as a function of interfacial electric field strength.

5. The claim that the energy of adsorbed H* is not affected by an interfacial electric field should be supported by DFT calculations.

Response

We now provide new DFT calculations on the *H adsorption energy on Ni-N₄, Fe-N₄ and Co-N₄ as a function of interfacial electric field strength (Supplementary Fig. 49, reproduced below). We observe that the adsorption energy of *H on Ni-N₄, Co-N₄ and Fe-N₄ does not change significantly with increasing electric field strength. Our results are consistent with a previous computational result (*J. Am. Chem. Soc.* 2017, 139, 32, 11277).

We have added the above results and discussion into the revised manuscript, which are also listed below:

*This is because the *H intermediate does not have a dipole moment or polarizability and hence is not affected by a change in the interfacial electric field. This is evident from our DFT calculations of the *H adsorption energy, which exhibited no significant changes as a function of interfacial electric field strength (Supplementary Fig. 49), and is consistent with prior literature^{29, 65}.*

Supplementary Fig. 49 | *H adsorption energy as a function of electric field strength on Ni-N₄, Fe-N₄, and Co-N₄.

6. The SHINERS technique used for measuring the interfacial electric field is novel and not well-established, according to the authors. It should be clearly explained in the text why this technique was chosen, and how SiO₂-coated gold particles were applied to the samples. It is also suggested to include the spectrum of the catalyst without adsorbed SCN⁻ as a control. Error bars should be added to the data in Figure 3d.

Response

The reason why we selected this technique is because without the enhancement effect of SHINERS, we were unable to detect any signal from the SCN⁻ peak. We postulate that this is because unlike bulk metallic surfaces, there are much less sites available for single-atom catalysts, making the Raman SCN⁻ peak naturally weaker. Hence, we found that we were only able to detect the SCN⁻ peak when we added the enhancement effect of SHINERS.

For SHINERS, 10 μ L (~0.5 mg/mL) of SHINERS was first dropped onto a glassy carbon electrode (3 mm in diameter). Following this, 5 μ L (5 mg/mL) of the catalyst ink was added. After the electrode was completely dried, it was immersed in 0.1 M NaClO₄ containing 0.1 M KSCN for 15 min and rinsed several times with deionized water before measurement.

In the methods section, we added:

For SHINERS, 10 μ L (~0.5 mg/mL) of SHINERS was first dropped onto a glassy carbon electrode (3 mm in diameter). Following this, 5 μ L (5 mg/mL) of the catalyst ink was added. After the electrode was completely dried, it was immersed in 0.1 M NaClO₄ containing 0.1 M KSCN for 15 min and rinsed several times with deionized water before measurement.

In addition, a control spectrum without introduction of SCN^- is provided in Supplementary Fig. 39 (reproduced below), where no SCN^- peak was observed as expected. Error bars are now included in Fig. 3d (reproduced below).

We have added the above results and discussion into the revised manuscript, which are also listed below:

Under normal in-situ Raman spectroscopy conditions, we were unable to detect the SCN^- peak. Hence, we employed shell-isolated nanoparticle-enhanced Raman spectroscopy (SHINERS) to enhance the Raman signal,⁵⁸ which uses SiO_2 -coated Au nanoparticles (see Supplementary Fig. 39 and methods section for more details).

Supplementary Fig. 39 | SHINERS enhanced Raman spectrum of Ni-SAC-250 without adsorbed SCN^- as a control.

Fig. 3 | (a) Electric field distribution on different size Ni-SACs deduced using Kelvin probe atomic force microscopy. (b) Schematic of probing the electric field of the active site in different sized Ni-SACs by using the vibrational stark effect. (c) Potential-dependent SHINERS spectra of SCN^- adsorbed on Ni-SAC-70 in 0.1 M NaClO_4 solution. (d) Peak positions of $\nu \text{SCN}^- \text{-NiN}_4$ as a function of the electrode potential.

7. The interfacial electric field increases the concentration of counter-ions in the vicinity of the electrode [Nature 537, no. 7620 (2016): 382-386]. These counter-ions can alter the adsorption energy of intermediates, which was not considered in the DFT simulations. This should be discussed in the text.

Response

As discussed in point 4, we have revised the DFT calculations by including K^+ ions and explicit solvent water into the model.

8. In Supplementary Note 1, the authors have mentioned the Electric Current Module for their simulation; it should be explained how this module was used for the electric field simulation that was performed, as the relevance is unclear.

Response

The Electric Current Module was used to set up the applied potential and to calculate the free electron density based on the equation:

$$\nabla \cdot D = \rho_V$$

This information has been added into Supplementary Note 1.

9. Statistical analysis of the nanoparticle size must be performed until the Gaussian shape of particle size distribution is achieved.

Response

We have reanalyzed the particle size distribution, and the revised distribution results are displayed in Fig. 2b-e and Supplementary Fig. 7, 10, 23, 24. The revised Fig. 2b-e is also listed below:

Fig. 2 | (b-e) TEM images of Ni SACs, consisting of Ni single-atoms supported on carbon nanospheres of varying diameter. (b) Ni-SAC-70, (c) Ni-SAC-130, (d) Ni-SAC-250 and (e) Ni-SAC-350. The inset of each image shows the size distribution of the carbon nanospheres. More TEM images are shown in Fig. S9.

Reviewer #3 (Remarks to the Author):

In this manuscript, the authors reported the preparation of single atom catalysts (SACs) consisting of M-N₄ sites anchored on a series of spherical hollow carbon supports with different diameters, and studied the effect of nanocurvature of carbon supports on electrocatalytic activities of SACs (CO₂RR, OER, and ORR). They try to attribute the activity modulation of SACs to nanocurvature induced interfacial electric field via theoretical calculations. However, there is a large discrepancy between the DFT models (flat surface) and the actual structure (curved surface) of the catalysts. The conclusions drawn from DFT calculations are contradictory with those of experimental results, in which the minimum curvature of carbon support corresponds to best performances of SACs in DFT calculations, while a volcanic performance trend is experimentally obtained. As a result, I am not convinced by the mechanism.

Response

We thank Reviewer #3 for taking the time to evaluate our manuscript and providing helpful suggestions to improve our manuscript.

The diameter of our investigated catalysts ranges from 70 nm (smallest) to 600 nm (largest). Under this sphere diameter range, the corresponding curvature in our DFT model with a size of only 20*30 Å means that it is essentially flat rather than a curved surface.

In response to the questions on the DFT calculations by Reviewer #2 and Reviewer #3, we have revised our simulation model and included K⁺ and H₂O into the simulation model. As shown in Fig. 1a (reproduced below), among the four elementary reaction steps, the formation of *COOH is calculated to be the rate-determining step (RDS). The free energy change in this step decreased from 0.96 eV to 0.88 eV when the electric field increased from 0 to 0.4 VÅ⁻¹. However, when the electric field was increased to -0.8 VÅ⁻¹, the free energy change in this step increased to 0.9 eV. This indicates there is an optimal electric field strength for CO₂R performance.

In addition, the limiting potential difference between CO₂R and HER (U_L(CO₂)-U_L(H₂); U_L= -ΔG₀/e) has also been calculated and employed as the descriptor of CO selectivity according to prior literature (*Angew. Chem. Int. Ed.* 2019, 58, 6972; *J. Am. Chem. Soc.* 2017, 139, 24, 8329), where a more positive value of U_L(CO₂)-U_L(H₂) represents a higher CO₂R selectivity. As shown in Fig. 1b, an optimal electric field strength was observed for this descriptor with Ni-N₄ and is qualitatively consistent with our experimental results.

We have added the above results and discussion into the revised manuscript, which are also listed below:

For example, the free energy change for this step (limiting potential) is reduced from 0.96 eV to 0.88 eV when an electric field of -0.4 VÅ⁻¹ is applied, but increased to 0.9 eV when the electric field was further increased to -0.8 VÅ⁻¹. In addition, the limiting potential difference between CO₂R

and HER ($U_L(\text{CO}_2)-U_L(\text{H}_2)$; $U_L = -\Delta G_0/e$) was also calculated and employed as the descriptor for CO selectivity^{45,46}, with a more positive value of $U_L(\text{CO}_2)-U_L(\text{H}_2)$ indicating a higher CO₂R selectivity. These findings are summarized in Fig. 1b, where an optimal electric field strength was observed for this descriptor with Ni-N₄.

The related references have been cited in the revised manuscript and are listed below:

- 45 Kim, D. et al. Electrochemical activation of CO₂ through atomic ordering transformations of AuCu nanoparticles. *J. Am. Chem. Soc.* **139**, 8329-8336 (2017).
 46 Ren, W. et al. Isolated diatomic Ni-Fe metal–nitrogen sites for synergistic electroreduction of CO₂. *Angew. Chem. Int. Ed.* **58**, 6972-6976 (2019).

Fig. 1 | (a) Reaction pathway Gibbs free energy diagrams and (b) theoretical limiting potential difference of $U_L(\text{CO}_2)-U_L(\text{H}_2)$ on Ni-N₄ as a function of interfacial electric field strength.

Some extra comments are listed below:

1. There are already several related reports concerning the curvature of supports modulated catalytic activities of SACs, such as Nat. Commun. 12, 6335 (2021), Angew. Chem. Int. Ed. 2021, 60, 22722–22728, Nat. Catal. (2023). <https://doi.org/10.1038/s41929-023-01005-3>, and ACS Energy Lett. 2023, 8, 1330–1335, which largely weaken the novelty of the current work.

Response

We have carefully reviewed the literature mentioned above with great interest and the related papers are now cited in our revised manuscript. We found that the above-mentioned literature mainly focuses on strain engineering and studying the effect of strain on the ORR and CO₂R performance. For instance, in (Nat. Catal. (2023). <https://doi.org/10.1038/s41929-023-01005-3>) the SWCNT diameter range investigated was (i) 4–6 nm, (ii) 5–15 nm, (iii) 10–20 nm, (iv) 20–30 nm and (v) >50 nm. From this, it is apparent that very small diameters are necessary to induce a strain effect on the active site. Importantly, this strain effect induces changes in the electronic structure of the active site, which the authors observed in their X-ray photoelectron spectroscopy (XPS) and X-ray absorption spectroscopy (XAS) measurements.

However, in our work, the investigated sphere diameter range is much larger (70-600 nm) which means the nanocurvature-induced strain effect is much weaker. More importantly, we do not observe any significant changes in our XPS (Supplementary Fig. 18, 35 and 36) and XAS (Supplementary Fig. 21 and 34) measurement results as a function of sphere diameter, which indicates that any nanocurvature-induced strain is negligible in our case.

We have added the above discussion into the revised manuscript, which is also listed below:

We also note that nanocurvature-induced strain effects⁵³⁻⁵⁶ are unlikely to operate in our investigated SAC systems, which requires a much higher curvature of the support. More importantly, our XPS and XAS results do not indicate any significant changes in the electronic structure of our catalysts as a function of sphere diameter, which indicates that any nanocurvature-induced strain is negligible.

The related references have been cited in the revised manuscript and are listed below:

- 53 Han, G. et al. Substrate strain tunes operando geometric distortion and oxygen reduction activity of CuN₂C₂ single-atom sites. *Nat. Commun.* **12**, 6335 (2021).
- 54 Yang, J. et al. Compressive strain modulation of single iron sites on helical carbon support boosts electrocatalytic oxygen reduction. *Angew. Chem. Int. Ed.* **60**, 22722-22728 (2021).
- 55 Su, J. et al. Strain enhances the activity of molecular electrocatalysts via carbon nanotube supports. *Nat. Catal.*, 1-11 (2023).
- 56 Cepitis, R., Kongi, N., Rossmeisl, J. & Ivaništšev, V. Surface curvature effect on dual-atom site oxygen electrocatalysis. *ACS Energy Lett.* **8**, 1330-1335 (2023).

2. It is important to elucidate the intermediate species and the mechanism of the interfacial electric field on the intermediate species, and thus in-situ Fourier transform infrared spectroscopy is necessary.

Response

We conducted the attenuated total reflectance surface-enhanced infrared absorption spectroscopy (ATR-SEIRAS) measurement on Ni-SACs with three different curvatures. The results as shown in Supplementary Fig. 41 (reproduced below). Peaks located at around 2353 cm⁻¹ and 1640 cm⁻¹ can be attributed to CO₂ absorption and stretching vibration of H₂O respectively. The peak at 1930 cm⁻¹ is a characteristic peak belonging to the stretching of a single metal site adsorbed CO (*Nat. Commun.* 2022, 13, 6082; *Nat. Commun.* 2023, 14, 3401). The accumulated metal-adsorbed CO could be detected over Ni-SAC-250, but was not observed on the Ni-SAC-130 and Ni-SAC-350 samples. Based on prior literature, this indicates that Ni-SAC-250 is better at converting CO₂ to CO (*Nat. Commun.* 2022, 13, 6082; *Angew. Chem. Int. Ed.* 2023, 62, e202215136). In addition, no apparent peak signal belonging to *COOH was observed in the spectrum of the three investigated catalysts. This might be attributed to the fact that acidic electrolyte was employed. This is consistent with a previously reported result (*Adv. Mater.* 2023, 35, 2209590), where *COOH was detected in neutral electrolyte but could not be detected in acidic electrolyte over the same catalyst.

We have added the above results and discussion into the revised manuscript, which are also listed below:

The in-situ Fourier Transform Infrared (FTIR) spectra of different diameter Ni-SACs also indicates that the Ni-SAC-250 catalyst is better at converting CO₂ to CO (Supplementary Fig. 41).

Supplementary Fig. 41 | *In-situ* FTIR spectra of Ni-SACs with different sphere diameter measured at $E = -0.4$ V to -1.5 V vs. Ag/AgCl in pH 1 acidic electrolyte (0.1 M K₂SO₄ + H₂SO₄). (a) Ni-SAC-130, (b) Ni-SAC-250 and (c) Ni-SAC-350. Peaks located at around 2353 cm⁻¹ and 1640 cm⁻¹ can be attributed to CO₂ absorption and stretching vibration of H₂O, respectively. The peak at 1930 cm⁻¹ is a characteristic peak belonging to the stretching of a single metal site adsorbed CO^{3,4}. The accumulated metal-adsorbed CO could be detected over Ni-SAC-250, but was not observed from the Ni-SAC-130 and Ni-SAC-350 samples, indicating that Ni-SAC-250 is better at converting CO₂ to CO^{3,5}. In addition, no apparent peak signal belonging to *COOH was observed in the spectrum of the three investigated catalysts. This might be attributed to the fact that acidic electrolyte was employed. This is consistent with a previously reported result, where *COOH was detected in neutral electrolyte but could not be detected in acidic electrolyte over the same catalyst⁶.

3. What are the thicknesses of spherical carbon supports and are they consistent? If there is a difference in thickness, what is the effect of thickness on catalytic performance?

Response

We have analyzed the shell thickness of the different diameter Ni-SAC catalysts and the results are shown in Supplementary Fig. 43 (reproduced below). Based on these results, we indeed found that catalysts with different sphere diameter have different shell thicknesses. Therefore, to understand if the shell thickness has a significant effect on the CO₂R performance, we synthesized a series of three catalysts with similar sphere diameters (250 nm) but different shell thickness by tuning the reaction temperature. As shown in Supplementary Fig. 44 (reproduced below), there is

no significant difference in the FE or TOFs for the three investigated Ni-SACs. This indicates that the shell thickness does not have a significant effect on the CO₂R performance. We reason that this is because our metal sites are located mainly on the surface of the carbon sphere (*Nat. Mater.*, 2021, 20(10), 1385-1391), hence the thickness of the shell does not have a significant impact.

We have added the above results and discussion into the revised manuscript, which are also listed below:

In addition, we found that catalysts of similar sphere diameter (250 nm) but different shell thickness still exhibit similar CO₂R performance. This allows us to rule out the impact of shell thickness on the catalytic activity (Supplementary Fig. 43, 44).

Supplementary Fig. 43 | Shell thickness analysis of Ni-SACs catalysts based on TEM images (a) Ni-SAC-70, (b) Ni-SAC-130, (c) Ni-SAC-250 and (d) Ni-SAC-350.

Supplementary Fig. 44 | (a-c) TEM image of Ni-SACs with similar diameter (250 nm) but different shell thicknesses and (d-f) their corresponding shell thickness analysis. (g) FE for CO production under different current densities and (h) TOFs at -1.5 V vs. RHE over the three catalysts with different shell thicknesses. ST stands for shell thickness.

4. For better contrast, the CO₂R performance of Fe-SACs and Co-SACs with diameters of 70, 130, 250 and 350 nm should also be provided.

Response

According to this suggestion, the CO₂R performance of Fe-SACs and Co-SACs with diameters of 70, 130, 250 and 350 nm have been evaluated and added into Fig. 4b and Fig. 4d (reproduced below) respectively.

Fig. 4 | FE for CO production over Ni-SACs (a), Fe-SACs (b) and Co-SACs (c) with different diameters as a function of applied current density measured in acidic electrolyte (0.1 M K₂SO₄ + H₂SO₄). The error bars represent standard deviation from three independent measurements. (d) TOFs for CO production at -1.5 V vs RHE over Ni-SACs, Fe-SACs, and Co-SACs as function of SAC sphere diameter. (e) Stability test of Ni-SAC-250 catalyst at 200 mA cm⁻² measured in acidic electrolyte.

5. Why is the DFT study of CO₂R focused on the Ni-N₄ SACs?

Response

Our DFT study of CO₂R focused on the Ni-N₄ system because it is the most widely reported active site for CO₂R to CO with single-atom catalysts. Furthermore, it also exhibited the best CO₂R performance among the 3 active sites investigated in our work (Ni-N₄, Fe-N₄ and Co-N₄). To further understand electric field effects on these other SACs for CO₂R, we performed new DFT calculations on Fe-N₄ and Co-N₄ as well. As shown in Supplementary Fig. 2d and 3d, a volcano plot of $(U_L(\text{CO}_2)-U_L(\text{H}_2))$ vs electric field strength can also be observed over the Fe-N₄ and Co-N₄ active sites respectively, similar to the case with Ni-N₄.

Supplementary Fig. 2 | Adsorption energies of CO₂R intermediates as a function of electric field strength on Fe-N₄. (a) *CO₂, (b) *COOH and (c) *CO. (d) theoretical limiting potential difference of $U_L(\text{CO}_2)-U_L(\text{H}_2)$ on Fe-N₄ under the influence of various interfacial electric field strengths.

Supplementary Fig. 3 | Adsorption energies of CO₂R intermediates as a function of electric field strength on Co-N₄. (a) *CO₂, (b) *COOH and (c) *CO. (d) theoretical limiting potential difference of U_L(CO₂)-U_L(H₂) on Co-N₄ under the influence of various interfacial electric field strengths.

6. Considering the differences in dimensions of different samples, the authors should standardize the scale in KPFM. Furthermore, there are some problems in the unit and legends, such as Fig. 3d and Fig. 5a.

Response

According to this suggestion, we attempted to standardize the KPFM data accordingly. However, since the sizes of the spheres are quite different, we found it difficult to display an entire carbon sphere for the largest diameter size when standardizing the scale of KPFM data. As a result, it becomes difficult to illustrate the field difference induced by the different sphere diameters (Fig. R1 below). Hence, we normalized the scale for the smallest two and also the largest two sized catalysts (Fig. 3a reproduced below). In addition, the problems of the unit and legends have been corrected in the revised manuscript.

Figure R1. Scale standardized KPFM image.

Fig. 3 | (a) Electric field distribution on different size Ni-SACs deduced using Kelvin probe atomic force microscopy. (b) Schematic of probing the electric field of the active site in different sized Ni-SACs by using the vibrational Stark effect. (c) Potential-dependent SHINERS spectra of SCN⁻ adsorbed on Ni-SAC-70 in 0.1 M NaClO₄ solution. (d) Peak positions of ν SCN-NiN₄ as a function of the electrode potential.

7. In Fig. 1a, the relative position of Gibbs free energy in the initial and final state is not correct. Please check them.

Response

The corresponding Gibbs free energy in the initial and final state have been corrected in the revised manuscript.

8. Some minor errors: Fig. 1d, line 50, VÅ-1 instead of eVÅ-1. Line 458, Table S3 instead of Table S1.

Response

These errors have been modified in our revised manuscript.

REVIEWER COMMENTS

Reviewer #1 (Remarks to the Author):

The authors have adequately addressed my concerns and I recommend this manuscript for publication in Nature Communications.

Reviewer #3 (Remarks to the Author):

The authors have addressed most of my concerns by adding the relevant experimental and theoretical data. However, the core mechanism behind the nanocurvature of carbon supports modulating electrocatalytic activities of SACs is still not convinced. As mentioned by the Reviewer#2, I highly agree that the experimental data in Figure 4a-d does not clearly support the DFT simulations and the effect of the curvature-enhanced electric field. In addition, it is hard to understand that the electronic structures of M-N4 atomic sites are not affected by the interfacial electric field induced by nanocurvature since electric field manipulating the electronic structures of active sites that have been widely reported (such as Nat. Commun. 12, 5128 (2021), Nat. Commun. 13, 3063 (2022), and J. Am. Chem. Soc. 144, 3039-3049 (2022)). Thus, extra comments are listed below:

1. Why K⁺ and H₂O were included in the simulation models? The previous work unveiled that adsorbed K⁺ ions on Au surface lower the thermodynamic energy barrier for CO₂ to CO reaction by stabilizing the intermediates (Nature 537, 382-386 (2016)). It is not clear that whether optimized reaction pathway of CO₂R to CO derives from the effect of K⁺ or electric field or their cooperation. If the electric field works, how does it regulate the adsorption of reaction intermediates and optimize reaction pathway? The electric field may modulate the electronic structures of active sites, such as d-band center, charge distribution, and spin state.
2. The authors believe that the electric field and the reaction intermediates together determine the catalytic reaction activity, revised DFT calculations by including K⁺ and H₂O into the simulation model. The local K⁺ concentration does affect the catalytic activity, but the authors do not discuss whether the introduction of the electric field cause the change in the local K⁺ concentration, and whether the effect of the electric field on K⁺ is crucial to determine on the catalytic activity. Furthermore, the authors take solvent effects into account in the calculation process, the details should be added in the experimental section.
3. The authors conclude that the metal sites are located mainly on the surface of the carbon spheres and that additional experimental evidence should be added.
4. The model structures should be given.

We thank the editor for handling our manuscript and the reviewers for their comments, which we have used to improve the quality of the work. Changes within the manuscript and supporting information are highlighted in yellow. Below is a point-by-point response to the reviewer comments, which are written in blue font.

Reviewer #1 (Remarks to the Author):

The authors have adequately addressed my concerns and I recommend this manuscript for publication in Nature Communications.

Response

We are thankful to Reviewer #1 for their positive evaluation.

Additional responses to rev #2:

1) Comment 1 from rev #2.

Response

We would like to clarify that the local chemical environment does indeed play a role in influencing the CO₂R performance and this is quite evident in our data. However, our main objective in these experiments was to show that despite the local chemical environment induced changes in CO₂R activity, the optimal sphere diameter with best CO₂R performance remains the same.

We first reproduce the original Reviewer #2 comments below:

“Based on the conditions used in this work, the mixed effects of counterions and the local chemical environment should be considered and discussed in the text [see e.g., Nature Catalysis 1.12 (2018): 946-951; Nature Catalysis 2.8 (2019): 702-708; Nature Catalysis 5.4 (2022): 268- 276]. Spheres with smaller radii pack differently compared to larger spheres, creating different diffusion rates and microenvironments. Therefore, the chemical environment is not the same around catalysts with different curvatures, which makes the electrochemical data convoluted and not unambiguously supportive of the claims of the paper. In other words, the experimental data supports the fact that the diameter of spherical particles affects the electrochemical performance, but the results are not justifiable solely based on the effect of the interfacial electric field on the energy of adsorbates to the exclusion of other factors at play, as they have not been ruled out.”

Based on this comment, we reasoned that varying the loading for each catalyst sphere diameter could serve to alter the local chemical environment. This is because each loading would be different in terms of packing, diffusion rates and microenvironments

(local pH and CO₂ availability). Our objective was therefore to alter the local chemical environment by varying the loadings to determine if the sphere diameter for optimal performance would remain the same or not. Importantly, if the sphere diameter for optimal performance is loading independent, then we can rule out the local chemical environment as a factor that controls the optimal sphere diameter for best performance.

The results for varying the catalyst loading are shown in Supplementary Figure 44. We observe that regardless of the catalyst loading, the optimal sphere diameter is 250 nm (Ni-SAC-250). Importantly, we can conclude that the local chemical environment does not control the optimal sphere diameter for the catalyst. Rather, this optimal sphere diameter is due to its optimal interfacial electric field strength for CO₂R.

Supplementary Fig. 44 | FE towards CO for Ni-SACs with different sphere diameters under various catalyst loadings (a) 0.2 mg cm⁻², (b) 0.5 mg cm⁻² and (c) 1.0 mg cm⁻² in acidic electrolyte (pH=1, H₂SO₄ + 0.1 M K₂SO₄ solution) at -1.5 V vs RHE.

However, we do acknowledge that even though the local chemical environment does not control the optimal sphere diameter, it does affect the observed CO₂ reduction activity. This can be seen in the contour plot in Fig. R1, where the FE to CO is plotted as a function of catalyst loading and catalyst sphere diameter. It is generally observed that higher loadings result in higher FE to CO and this is consistent with prior literature results (*Science* 2019, 364(6445), 1091-1094) (*Chem. Sci.* 2021, 12(47), 15682-15690). Hence, the local chemical environment clearly has an effect on the observed CO₂R activity, however this does not control the optimal sphere diameter. Instead, this optimal sphere diameter is because its nanocurvature gives rise to an optimal interfacial electric field for promoting CO₂R as discussed in our main text.

Regarding this issue, we have included more discussion in the main text to improve clarity and understanding.

Fig. R1 | Contour plot showing the FE towards CO for Ni-SACs as a function of sphere diameter and catalyst loading. CO₂R was performed in acidic electrolyte (pH=1, H₂SO₄ + 0.1 M K₂SO₄ solution) at -1.5 V vs RHE.

2) Comment 7 from rev #2.

Response

We conducted DFT calculations on Ni-N₄ both in the presence and absence of K⁺ counter-ions and without an electric field. As shown in Supplementary Fig. 1, we find that the addition of K⁺ does indeed enable a reduction in the free energy change for the potential limiting step for CO₂RR to CO. We note that this result is consistent with the prior literature (*Nature* 2016, 537, 382-38). Cations have been proposed to enhance CO₂R activity because these can influence interfacial electric fields, which then changes intermediate adsorption energies and modifies the catalytic activity (*Nat. Catal.* 2022, 5, 268–276) (*Energy Environ. Sci.* 2019,12, 3001-3014) (*J. Am. Chem. Soc.* 2017, 139, 32, 11277–11287).

Supplementary Fig. 1 | Reaction pathway Gibbs free energy diagrams of Ni-N₄ in the (a) presence of K⁺ and (b) absence of K⁺. The free energy change of the rate-limiting step is shown in each case. Structure models of Ni-N₄ with K⁺ and H₂O adsorbed with the intermediates: (c) *CO₂, (d) *COOH, and (e) *CO. The brown, gray, blue, red, purple, and light pink spheres represent carbon, nitrogen, nickel, oxygen, potassium, and hydrogen atoms respectively.

To understand whether changing the sphere diameter affects the local K⁺ concentration near the catalyst surface, we used a finite-element numerical method to quantify the surface K⁺ concentration of different diameter catalysts. The result in Supplementary Fig. 8 shows that as the sphere diameter decreases from 1000 nm to 100 nm, the surface K⁺ concentration increases from 0.219 to 1.644 M. This result is also consistent with the prior literature (*Nature* 2016, 537, 382-386), where increases in the interfacial electric field results in a corresponding increase in the surface K⁺ concentration. This result can be understood based on electrostatics. Since K⁺ has a positive charge, it serves as a “counter-ion” that moves closer to the electrode surface in response to an increasing interfacial electric field.

In addition, the positive charge on K⁺ means that the electrostatic effect experienced by adsorbed intermediates is a combined contribution of the electrode (negatively charged) and these counter-ions (positively charged). This was previously depicted by Bell and co-workers in a schematic (Fig. R2), where the authors investigated these effects both experimentally and theoretically for CO₂R on both Cu and Ag surfaces (*J. Am. Chem. Soc.* 2017, 139, 32, 11277–11287). They found that these interfacial electric fields have

a significant influence on the adsorption energies of intermediates which then strongly impacts the catalytic activity.

Supplementary Fig. 8 | Finite-element numerical method simulation results of the local surface K⁺ concentration as a function of sphere diameter. Bulk electrolyte K⁺ concentration was set at 0.2 M.

Fig. R2 | Depiction of the interfacial electric field at the electrode surface by Bell and co-workers (*J. Am. Chem. Soc.* 2017, 139, 32, 11277–11287). The electrostatic effect experienced by adsorbed intermediates is a combined contribution from the electrode (brown spheres) and the cation counter-ion (purple sphere).

In summary, we can conclude that the sphere diameter (nanocurvature) of our single-atom catalyst system can be used as a knob to modify the interfacial electric field. This change of interfacial electric field with sphere diameter was evidenced by our finite-element numerical method simulations (Fig. 1e, f), Kelvin probe force microscopy (Fig. 3a) and *in-situ* Raman spectroscopy results (Fig. 3c-d). The stronger interfacial electric field also results in a higher concentration of K⁺ at the surface. Due to the positive charge on K⁺, the electrostatic effect experienced by adsorbed intermediates is a combination of the counter-ions (positively charged) and the electrode (negatively

charged). This modifies adsorbate binding energies based on their polarizability and dipole moment and can be given by this equation (*J. Phys. Chem. C* 2020, 124, 14581–14591) below:

$$G_{ads} = G_{ads}^{PZC} + \vec{\mu} \cdot \vec{E} - \frac{\alpha}{2} \vec{E}^2$$

Where G_{ads} refers to the adsorbate binding energy, G_{ads}^{PZC} refers to the adsorbate binding energy with no applied interfacial electric field, $\vec{\mu}$ is the adsorbate dipole moment, α is the adsorbate polarizability and \vec{E} is the interfacial electric field.

Hence, these offer an explanation why controlling the sphere diameter (nanocurvature) impacts the activity of our catalysts. These discussion points have been added to the manuscript.

Reviewer #3 (Remarks to the Author):

The authors have addressed most of my concerns by adding the relevant experimental and theoretical data.

Response

We are thankful for the reviewer's positive comments. Based on the reviewer's helpful suggestions, we have improved the discussion on the role of K^+ cations on impacting the interfacial electric field.

However, the core mechanism behind the nanocurvature of carbon supports modulating electrocatalytic activities of SACs is still not convinced. As mentioned by the Reviewer#2, I highly agree that the experimental data in Figure 4a-d does not clearly support the DFT simulations and the effect of the curvature-enhanced electric field.

Response

In the previous version, we reconducted the DFT simulations by adding K^+ and H_2O into the simulation models. As shown in Figure 1a, among the four elementary reaction steps, the formation of $*COOH$ is calculated to be the rate-determining step (RDS). The energy barrier decreased from 0.96 eV to 0.88 eV when the electric field increased from 0 to 0.4 VA^{-1} , but further increasing the electric field to -0.8 VA^{-1} , the energy barrier increased to 0.9 eV, indicating there is an optimal electric field strength for the catalyst. In addition, the limiting potential difference between CO_2R and HER ($U_L(CO_2) - U_L(H_2)$; $U_L = -\Delta G_0/e$) has been further calculated and employed as the descriptor of CO selectivity, where a more positive value of $U_L(CO_2) - U_L(H_2)$ represents a higher CO_2R selectivity (*Angew. Chem. Int. Ed.* 2019, 58, 6972) (*J. Am. Chem. Soc.* 2017, 139, 24, 8329). As shown in Figure 1b, a volcano plot was observed over the Ni- N_4 under different electric field strengths, consistent with our experimental results.

Fig. 1 | (a) Reaction pathway Gibbs free energy diagrams and (b) theoretical limiting potential difference of $U_L(\text{CO}_2) - U_L(\text{H}_2)$ on Ni-N₄ under the influence of various interfacial electric field strengths.

In addition, it is hard to understand that the electronic structures of M-N₄ atomic sites are not affected by the interfacial electric field induced by nanocurvature since electric field manipulating the electronic structures of active sites that have been widely reported (such as Nat. Commun. 12, 5128 (2021), Nat. Commun. 13, 3063 (2022), and J. Am. Chem. Soc. 144, 3039-3049 (2022)).

Response

The work by Ju et al. (Nat. Commun. 12, 5128 (2021)) theoretically investigated the transition metal single atoms anchored onto α -In₂Se₃ monolayers as ferroelectric SACs for electrochemical CO₂ reduction. They found that switching the polarization of these SACs can alter the reaction barrier or paths of CO₂ reduction, thus improving the CO₂ reduction performance. They also claim that these performance improvements stem from the synergistic effects of adjusted empty and occupied d-orbitals (d orbital center) of adsorbed metal atoms, polarization-dependent electron transfer, and CO₂ adsorption energies under ferroelectric switching. We found that this work mainly studied the effect of ferroelectric polarization of these α -In₂Se₃-based SACs for CO₂ reduction. Hence, we feel that this may not be so relevant to our work, since we instead focus on the interfacial electric field.

The work by Pan et al. (Nat. Commun. 13, 3063 (2022)) is related to our work and is now cited in our manuscript. They demonstrated that an externally applied oriented electric field was able to enhance the HER performance of 2D SAC, such as Pt SAs-MoS₂ and Co SAs-WSe₂. They revealed that these external electric fields have a significant influence on the charge redistribution of SACs (especially at the active sites) and alter the kinetics of the rate-determining steps, thus leading to enhanced catalytic activity. We find that our main conclusions are consistent with this work, especially regarding the notion that the applied potential can change intermediate adsorption

energies and alter the kinetics of the rate-determining steps, which can, in turn, significantly impact the reaction mechanisms.

The work by Yang et al. (*J. Am. Chem. Soc.* 144, 3039-3049 (2022)) was previously cited and discussed in the introduction of our manuscript. This work reported a conformal coating method to simultaneously enhance the local electric field and temperature of the Cu nanoneedle simultaneously. They found that the enhanced local electric field and temperature synergistically promote the C-C coupling in CO₂ reduction. However, the authors did not investigate how the electric field affected the electronic structures of active sites in this work.

Overall, we find that the works by Pan et al. (*Nat. Commun.* 13, 3063 (2022)) and Yang et al. (*J. Am. Chem. Soc.* 144, 3039-3049 (2022)) are consistent with our work, that the electric field can affect the activity of the catalyst.

We emphasize that the main focus of our work is demonstrating that the sphere diameter (nanocurvature) can be used to control the interfacial electric field, which then influences the catalytic activity of single-atom catalysts. Notably, this change of interfacial electric field with sphere diameter was supported by our finite-element numerical method simulations (Fig. 1e, f), KPFM (Fig. 3a) and *in-situ* Raman spectroscopy results (Fig. 3c-d).

With regards as to how the interfacial electric field affects the catalytic activity, there are many proposed mechanisms in the literature as the reviewer correctly points out. This includes electronic structures of active sites, such as d-band center, charge distribution, spin state, concentration of cations and altering of intermediate adsorption energies. We note that explaining how the interfacial electric field influences the catalytic activity is still an open question for exciting future research, but we emphasize that this is not the core focus of our manuscript.

The reason why we used interfacial electric field modification of adsorption energies to explain our results was because this has been previously proposed by several research groups. For instance, Norskov and co-workers (*J. Phys. Chem. C* 2020, 124, 27, 14581-14591) used DFT simulations to investigate how the interfacial electric field affects the intermediate adsorption energies for the oxygen reduction reaction. Bell and co-workers adopted a similar approach (*J. Am. Chem. Soc.* 2017, 139, 32, 11277–11287) to investigate how interfacial electric fields affect the adsorption energies for the CO₂ reduction reaction. Similarly, Chan and co-workers (*Energy Environ. Sci.* 2019,12, 3001-3014) showed that cations impact the interfacial electric field, which then modifies intermediate adsorption energies and impacts the catalytic activity.

These discussion points are now included in the manuscript.

Thus, extra comments are listed below:

1a. Why K⁺ and H₂O were included in the simulation models?

Response

The inclusion of K⁺ and H₂O inclusion into the model was also performed in the work by Liu and co-workers (*J. Am. Chem. Soc.* 2022, 144, 7, 3039–3049). In addition, Reviewer #2 had suggested we include cations into our DFT simulations. Hence, for these reasons we decided to include them.

Below we reproduce the original comment by Reviewer #2:

“The interfacial electric field increases the concentration of counter-ions in the vicinity of the electrode [Nature 537, no. 7620 (2016): 382-386]. These counter-ions can alter the adsorption energy of intermediates, which was not considered in the DFT simulations. This should be discussed in the text.”

Based on this comment, we agreed with Reviewer #2 that counter-ions (e.g. K⁺) can alter the adsorption energy of intermediates. Such cations have been proposed to enhance CO₂R activity because these can influence interfacial electric fields, which then changes intermediate adsorption energies and modifies the catalytic activity (*Nat. Catal.* 2022, 5, 268–276) (*Energy Environ. Sci.* 2019,12, 3001-3014) (*J. Am. Chem. Soc.* 2017, 139, 32, 11277–11287).

1b. The previous work unveiled that adsorbed K⁺ ions on Au surface lower the thermodynamic energy barrier for CO₂ to CO reaction by stabilizing the intermediates (Nature 537, 382-386 (2016)). It is not clear that whether optimized reaction pathway of CO₂R to CO derives from the effect of K⁺ or electric field or their cooperation. If the electric field works, how does it regulate the adsorption of reaction intermediates and optimize reaction pathway? The electric field may modulate the electronic structures of active sites, such as d-band center, charge distribution, and spin state.

Response

We conducted DFT calculations on Ni-N₄ both in the presence and absence of K⁺ counter-ions. As shown in Supplementary Fig. 1, we find that the addition of K⁺ does indeed enable a reduction in the free energy change for the potential limiting step for CO₂RR to CO. We note that this result is consistent with the prior literature (*Nature* 2016, 537, 382-386) as the reviewer has mentioned. Cations have been proposed to enhance CO₂R activity because their positive charges can influence interfacial electric fields, which then changes intermediate adsorption energies and modifies the catalytic activity (*Nat. Catal.* 2022, 5, 268–276) (*Energy Environ. Sci.* 2019,12, 3001-3014) (*J. Am. Chem. Soc.* 2017, 139, 32, 11277–11287).

Supplementary Fig. 1 | Reaction pathway Gibbs free energy diagrams of Ni-N₄ in the (a) presence of K⁺ and (b) absence of K⁺. The free energy change of the rate-limiting step is shown in each case. Structure models of Ni-N₄ with K⁺ and H₂O adsorbed with the intermediates: (c) *CO₂, (d) *COOH, and (e) *CO. The brown, gray, blue, red, purple, and light pink spheres represent carbon, nitrogen, nickel, oxygen, potassium, and hydrogen atoms respectively.

To understand whether changing the sphere diameter affects the local K⁺ concentration near the catalyst surface, we used a finite-element numerical method to quantify the surface K⁺ concentration of different diameter catalysts. The result in Supplementary Fig. 8 shows that as the sphere diameter decreases from 1000 nm to 100 nm, the surface K⁺ concentration increases from 0.219 to 1.644 M. This result is also consistent with the prior literature (*Nature* 2016, 537, 382-38), where increases in the interfacial electric field results in a corresponding increase in the surface K⁺ concentration. This result can be understood based on electrostatics. Since K⁺ has a positive charge, it serves as a “counter-ion” that moves closer to the electrode surface in response to an increasing interfacial electric field.

In addition, the positive charge on K⁺ means that the electrostatic effect experienced by adsorbed intermediates is a combined contribution of the electrode (negatively charged) and these counter-ions (positively charged). This was previously depicted by Bell and co-workers in a schematic (Fig. R2), where the authors investigated these effects both experimentally and theoretically for CO₂R on both Cu and Ag surfaces (*J. Am. Chem. Soc.* 2017, 139, 32, 11277–11287). They found that these interfacial electric fields have

a significant influence on the adsorption energies of intermediates which then strongly impacts the catalytic activity.

Supplementary Fig. 8 | Finite-element numerical method simulation results of the local surface K⁺ concentration as a function of sphere diameter. Bulk electrolyte K⁺ concentration was set at 0.2 M.

Fig. R2 | Depiction of the interfacial electric field at the electrode surface by Bell and co-workers (*J. Am. Chem. Soc.* 2017, 139, 32, 11277–11287). The electrostatic effect experienced by adsorbed intermediates is a combined contribution from the electrode (brown spheres) and the cation counter-ion (purple sphere).

Hence, we can conclude that the sphere diameter (nanocurvature) of our single-atom catalyst system can be used as a knob to modify the interfacial electric field. This change of interfacial electric field with sphere diameter was evidenced by our finite-element numerical method simulations (Fig. 1e, f), Kelvin probe force microscopy (Fig. 3a) and *in-situ* Raman spectroscopy results (Fig. 3c-d). The stronger interfacial electric field also results in a higher concentration of K⁺ at the surface. Due to the positive charge on K⁺, the electrostatic effect experienced by adsorbed intermediates is a

combination of the counter-ions (positively charged) and the electrode (negatively charged). This modifies adsorbate binding energies based on their polarizability and dipole moment and can be given by this equation (*J. Phys. Chem. C* 2020, 124, 14581–14591) below:

$$G_{ads} = G_{ads}^{PZC} + \vec{\mu} \cdot \vec{E} - \frac{\alpha}{2} \vec{E}^2$$

Where G_{ads} refers to the adsorbate binding energy, G_{ads}^{PZC} refers to the adsorbate binding energy with no applied interfacial electric field, $\vec{\mu}$ is the adsorbate dipole moment, α is the adsorbate polarizability and \vec{E} is the interfacial electric field.

Hence, these offer an explanation on why controlling the sphere diameter (nanocurvature) impacts the activity of our catalysts.

With regards as to other means by which the interfacial electric field affects the catalytic activity, there are many proposed mechanisms in the literature as the reviewer correctly pointed out. This includes electronic structures of active sites, such as d-band center, charge distribution and spin state. We note that explaining how the interfacial electric field influences the catalytic activity is still an open question for exciting future research, but we emphasize that this is not the core focus of our manuscript. The core focus of manuscript is on demonstrating that the sphere diameter (nanocurvature) can be used to modify interfacial electric fields, which then impacts the catalytic activity.

The reason why we used interfacial electric field modification of adsorption energies as the most likely explanation was because this has been previously proposed by several research groups. For instance, Norskov and co-workers (*J. Phys. Chem. C* 2020, 124, 27, 14581–14591) used DFT simulations to investigate how the interfacial electric field affects the intermediate adsorption energies for the oxygen reduction reaction. Bell and co-workers adopted a similar approach (*J. Am. Chem. Soc.* 2017, 139, 32, 11277–11287) to investigate how interfacial electric fields affect the adsorption energies for the CO₂ reduction reaction. Similarly, Chan and co-workers (*Energy Environ. Sci.* 2019, 12, 3001-3014) showed that cations impact the interfacial electric field, which then modifies intermediate adsorption energies and impacts the catalytic activity.

These discussion points have been added to the manuscript.

2a. The authors believe that the electric field and the reaction intermediates together determine the catalytic reaction activity, revised DFT calculations by including K⁺ and H₂O into the simulation model. The local K⁺ concentration does affect the catalytic activity, but the authors do not discuss whether the introduction of the electric field cause the change in the local K⁺ concentration, and whether the effect of the electric field on K⁺ is crucial to determine on the catalytic activity.

Response

Please see the response to the previous comment.

2b. Furthermore, the authors take solvent effects into account in the calculation process, the details should be added in the experimental section.

Response

We have added the relevant details to the experimental section and these also provided below.

For all simulation models, one layer of water molecules was added to the surface to take solvation into account. Specifically, one potassium cation and six water molecules were added near to the surface (see model structures in Supplementary Figure S2). Additionally, the binding energy was calculated from DFT-optimized structures as follows: $E_{\text{binding}} = E_{\text{CO}_2^*} - (E_{\text{slab}} + E_{\text{CO}_2})$ where $E_{\text{CO}_2^*}$ is the energy of the system with CO_2 proximate to the slab surface, E_{slab} is the energy of the M-N₄ surface (with and without K⁺ and six water molecules for the respective cases), and E_{CO_2} is the gas-phase energy of CO_2 .

3. The authors conclude that the metal sites are located mainly on the surface of the carbon spheres and that additional experimental evidence should be added.

Response

We agree that it may be difficult to completely ascertain that all sites are located on the surface, hence we have removed this claim from the manuscript.

However, we were interested to assess the number of surface-exposed metal sites. Hence, to probe the surface density of the Ni-N₄ site in the samples with different shell thickness, we conducted cryo CO-pulse chemisorption measurement, which has been previously used to quantify the number of surface-exposed metal sites for M-N-C catalysts (*Nat. Commun.* 2015, 6(1), 8618) (*Nat. Mater.* 2020, 19(11), 1215). Two samples with a shell thickness (ST) of ST-24 (thinnest) and ST-40 nm (thickest) were tested. The results are as shown in Fig. R3 (reproduced below), where we found that the CO uptake amount is quite similar in the two samples. This indicates that the surface-exposed metal sites in both cases are quite similar. Hence, this provides support that the shell thickness is not a factor that affects our catalysis.

Fig. R3 | (a) Carbon monoxide pulse chemisorption profiles. (b) Moles of carbon monoxide that chemisorbed per gram of catalyst.

4. The model structures should be given.

Response

The model structures used in our calculations have been added as Supplementary Fig. 52 and are also reproduced below:

Supplementary Fig. 52 | Model structures of NiN₄, FeN₄ and CoN₄ with K⁺ and H₂O absorbed with different CO₂R intermediates. The brown, gray, blue, yellow, dark blue, red, purple, and light pink spheres represent carbon, nitrogen, nickel, iron, cobalt, oxygen, potassium, and hydrogen atoms, respectively.

Once again, we would like to sincerely thank the reviewer for taking the time to evaluate our manuscript and helping to improve the quality of our work.

REVIEWERS' COMMENTS

Reviewer #3 (Remarks to the Author):

The concerns from the reviewer have been addressed, and thus this manuscript can be considered to be accepted for publication.